# Super Consistency of Neural Network Landscapes and Learning Rate Transfer

**Lorenzo Noci**[*1]    **Alexandru Meterez**[*3 4 5]    **Thomas Hofmann**[1]    **Antonio Orvieto**[2 3 4]

## Abstract

Recently, there has been growing evidence that if the width and depth of a neural network are scaled toward the so-called rich feature learning limit ($\mu$P and its depth extension), then some hyperparameters — such as the learning rate — exhibit transfer from small to very large models. From an optimization perspective, this phenomenon is puzzling, as it implies that the loss landscape is consistently similar across very different model sizes. In this work, we study the landscape through the lens of the loss Hessian, with a focus on its largest eigenvalue (i.e. the sharpness), and find that certain spectral properties under $\mu$P are largely independent of the size of the network, and remain consistent as training progresses. We name this property *Super Consistency* of the landscape. On the other hand, we show that in the Neural Tangent Kernel (NTK) and other scaling regimes, the sharpness exhibits very different dynamics at different scales. But what causes these differences in the sharpness dynamics? Through a connection between the Hessian's and the NTK's spectrum, we argue that the cause lies in the presence (for $\mu$P) or progressive absence (for the NTK scaling) of feature learning. We corroborate our claims with a substantial suite of experiments, covering a wide range of datasets and architectures: from ResNets and Vision Transformers trained on benchmark vision datasets to Transformers-based language models trained on WikiText.

## 1   Introduction

Recent trends in deep learning research have unmistakably shifted towards an increase in model sizes, with networks comprising of billions of parameters emerging as the standard [1]. However, as models enlarge, so does the cost incurred in hyperparameter tuning which has led researchers to look for ways to scale up the architecture — both in terms of width and depth — while preserving the optimal hyperparameters (such as the learning rate).

While there exist several ways (a.k.a *parametrizations*) to scale up the width and depth of the network, not all of them facilitate learning rate transfer. For standard deep learning practices, such as networks parametrized with LeCun/Kaiming initializations [2, 3], a significant shift in the optimal learning rate is usually observed as the width and the depth of the model are increased. Similarly, under the Neural Tangent Kernel (NTK) parametrization [4], which provides theoretical insights into the behavior of very wide neural networks during training, the optimal learning rate also varies as the width and depth of the network change. Alternatively, Yang and Hu [5] and Yang et al. [6] propose the $\mu$P framework, designed to maximize the gradient update of the representations of the intermediate layers (i.e. feature learning) as the width increases. Under $\mu$P scaling, and its depth extension for residual networks Depth-$\mu$P [7, 8], it has been empirically demonstrated that the learning

---

[*]: Equal contribution. Correspondence to: `lorenzo.noci@inf.ethz.ch, ameterez@fas.harvard.edu`
[1]ETH Zürich, [2]ELLIS Tübingen, [3]MPI for Intelligent Systems, [4]Tübingen AI Center, [5]Harvard University

38th Conference on Neural Information Processing Systems (NeurIPS 2024).

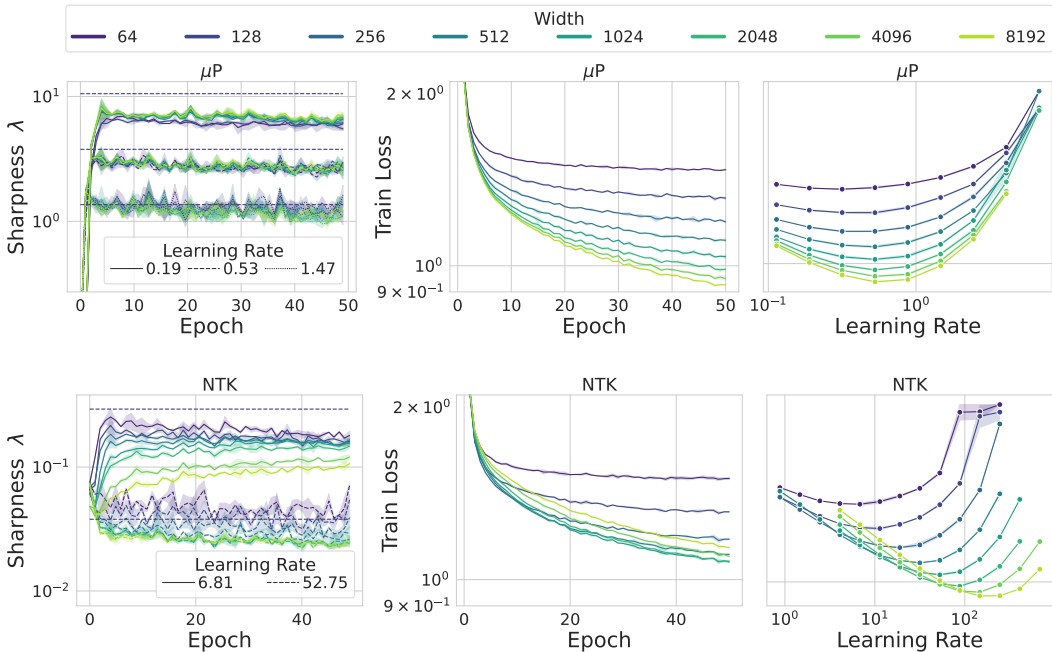

Figure 1: **Top row**. Under $\mu$P, (left) the sharpness dynamics are largely identical for the whole training dynamics across different widths, phenomenon that we call *Super Consistency*. The dashed horizontal lines are the Edge of Stability thresholds. Center: The loss dynamics are similar early in training, but accumulate finite-size effects over time, thus violating Super Consistency. Right: The learning rate transfers from small to large model, suggesting that the loss landscape is Super Consistent across different model sizes. **Bottom row**. Under NTK parameterization (NTP), the sharpness dynamics show large discrepancies. Also, the learning rate does not transfer. The architecture is a two-layer convolutional network trained on CIFAR-10 with data augmentation, where the width corresponds to the number of filters in the convolution. (See App. J). Other parameters: $B = 128$, epochs $= 50$.

rate transfers across both width and depth. In Vyas et al. [9] it is observed that in feature learning parametrizations (e.g. $\mu$P) the model's dynamics are *consistent* across model sizes, but for harder tasks or longer training times there are progressive and significant deviations across different model sizes. We give an example in Figure 1 (top center), where the training losses exhibits an increasing gap. The fact that the learning rate is exactly preserved, however, suggests that some properties of the landscape do not exhibit these finite-size deviations, and must be precisely preserved across different model sizes for the whole training trajectory.

Motivated by this, in the present work we identify the notion of *Super Consistency* to describe the properties of the neural network's loss landscape that are preserved across training as a function of the model width and depth, thus not accumulating finite-size effects. In particular, we analyze the landscape through the lens of the loss Hessian. It provides insights into the landscape's local curvature, and its structure for neural networks has been studied in several works [10–14]. Of great interest in optimization theory is the *sharpness*, i.e. the top Hessian eigenvalue, which for neural networks exhibit a rapid increase (*progressive sharpening*) towards a threshold called Edge of Stability (*EoS*) [15, 16]. However, although a few works have provided early insights [10, 16, 9], the scaling properties of the sharpness and Hessian's dynamics under different scaling limits remain unexplored. In this work we first present evidence of Super Consistency in the Hessian's largest eigenvalues, which have been shown to control the curvature along the optimization subspace [17]. We then focus on the sharpness dynamics, and find that the presence (resp. absence) of Super Consistency correlates well with presence (resp. absence) of learning rate transfer under $\mu$P, NTP and other scaling limits. These results suggest that learning rate transfer happens in super consistent landscapes, as the geometry of the landscape does not significantly change with the network's size.

Then, we investigate the role of feature learning in the progressive sharpening phase, and argue that while in $\mu$P feature learning causes progressive sharpening to reach a width-independent sharpness,

in the NTK regime the progressive lack of feature learning when the width is increased prevents the Hessian from adapting, and its largest eigenvalue from reaching the convergence threshold.

More concretely:

- We define Super Consistency, and show that under $\mu$P and Depth-$\mu$P it holds for the largest eigenvalues of the loss Hessian (Fig. 2), which converge to a largely width-independent threshold and remains there for the rest of training. On the other hand, we show that other quantities, such as the training loss and the NTK eigenvalues accumulate significant finite-size effects. We quantify the rate of divergence of these quantities with power law fits (Fig. 3).

- We analyze the relationship between Super Consistency of the sharpness and learning rate transfer across $\mu$P, Depth-$\mu$P, NTP and other parametrizations (Fig. 1, Fig. 4 and Sec. B). For $\mu$P and Depth-$\mu$P, which do transfer, the sharpness stays super consistent, stabilizing to a threshold (Fig. 1, top left), which in some cases corresponds Edge of Stability [16], and oscillates around it for a sustained period of training time. On the other hand, under NTP, *Standard Parametrization* (SP), or Depth-$\mu$P with multiple layers per residual block, the sharpness dynamics significantly separate during training for different widths, albeit in different ways. Also, here we do not observe transfer.

- We reproduce some of these results at realistic scale, including ResNets and Vision Transformers (ViTs) trained on Imagenet and GPT-2 on text data. Also, we analyze the effect of batch size, learning rate warm-up, and long training times.

- In Sec. 4.1 we show that the progressive sharpening phase is mainly driven by the NTK's largest eigenvalue, which is asymptotically fixed to its initial value for NTP, while it evolves at any width under $\mu$P. In Sec. 5 we provide intuition with a theoretical analysis on a two-layer linear network.

Finally, in Sec. 6 we discuss to what extent Super Consistency of these properties explains learning rate transfer, and the relevance of our results in the existing literature on optimization and scaling limits. Due to page limitations, we defer the discussion on related work to the appendix (App. A).

## 2    Background and Definitions

We consider a neural network with residual connections, defined by the following recursive equations over the layer indexes $\ell \in [L]$:

$$h^{\ell+1}(x) = \tau h^\ell(x) + \frac{1}{\sqrt{N}L^\alpha}W^\ell \phi(h^\ell(x)), \tag{1}$$

where $N$ and $L$ are the width and depth of the network, $W^\ell \in \mathbb{R}^{N \times N}$ for $\ell = 1, \ldots, L-1$, and $\tau$ is a factor that enables ($\tau = 1$) or disables ($\tau = 0$) the skip branch. We denote the output with $f(x) = \frac{1}{\gamma}W^L \phi(h^L(x))$, where $W^L \in \mathbb{R}^{1 \times N}$ and $\gamma$ scales the network output. Similarly, $\alpha$ has the role of interpolating between different depth limit regimes. At the first layer, we define $h^1(x) = \frac{1}{\sqrt{D}}W^0 x$, where $W^0 \in \mathbb{R}^{N \times D}$. All the weights $\theta = \{W^\ell\}_{l=0}^L$ are initialized independently from $\mathcal{N}(0,1)$ and we denote with $P$ the total number of parameters. We stress that the fully connected layer can be replaced with any type of layer (our experiments include convolutional and attention layers). Given a dataset $\mathcal{D} = \{(x_\mu, y_\mu)\}_{\mu=1}^{|\mathcal{D}|}$ of datapoints $x_\mu \in \mathbb{R}^D$ and labels $y_\mu \in \mathbb{R}$, we train the network with stochastic gradient descent (SGD) with batch size $B$ and learning rate $\eta \in \mathbb{R}$,

$$\theta_{t+1} = \theta_t - \eta \sum_{\mu=1}^B \nabla_\theta \mathcal{L}(f_t(x_\mu)), \tag{2}$$

where the loss $\mathcal{L}$ is a twice differentiable loss function. Defining $f_t := (f_t(x_\mu))_{\mu \in [|\mathcal{D}|]} \in \mathbb{R}^{|\mathcal{D}|}$ to be the vector of network's outputs at time $t$, if one considers continuous time, the corresponding gradient descent dynamics in function space $df_t/dt$ take the following form [18]: $\frac{df_t}{dt} = -\Theta(f_t)\Delta(f_t)$, where $\Delta(f_t)_i := \partial \mathcal{L}(f_t(x_i))/\partial f_t(x_i)$, $i \in [|\mathcal{D}|]$ is the vector of residuals, and $\Theta(f_t)_{ij} := \langle \nabla_\theta f_t(x_i), \nabla_\theta f_t(x_j) \rangle$ for $i, j \in [|\mathcal{D}|]$ is the Neural Tangent Kernel (NTK).

**Infinite Width.** The parameters $\gamma, \alpha, \eta \in \mathbb{R}$ determine the nature of the scaling limit. If $\gamma = \gamma_0, \eta = \eta_0$ are $\mathcal{O}(1)$ constants with respect to $N, L$ (neural tangent parameterization, or NTP), then the network enters the NTK regime [4]. Here, in the limit of infinite width, the NTK remains constant to its value at initialization throughout training, i.e. $\Theta(f_t) = \Theta(f_0)$ for all $t \geq 0$. Thus,

the network's dynamics become equivalent to a linear model trained on the first order term of the Taylor expansion of the model at initialization [19]. The fact that the NTK is fixed to its value at initialization is associated with the lack of feature learning of the model in the large width limit. If $\gamma = \gamma_0 \sqrt{N}$, and $\eta = \eta_0 \gamma^2$ ($\mu$P, or mean-field parameterization), the features evolve in the limit (i.e. the NTK $\Theta(f_t)$ evolves), and the richer model's dynamics can be described using either Tensor Programs [5] or dynamical mean field theory [20]. Under $\mu$P, Yang et al. [6] show that the learning rate $\eta_0$ as well as other hyperparameters transfer across width, in contrast to kernel limits, which we reproduce for our residual network in Fig. 1.

**Infinite Depth.** If on top of the $\mu$P framework, the residual branches are scaled with $\alpha = 1/2$ (Depth-$\mu$P), then Bordelon et al. [7] and Yang et al. [8] show that the infinite width dynamics also admit a feature-learning infinite depth limit. Under Depth-$\mu$P, the learning rate $\eta_0$ transfers with both width and depth. In this paper, we compare NTP and $\mu$P regimes as the width is increased, and show that our results extend to depth-scaling using the Depth-$\mu$P model. We summarize the feature learning parametrizations and report Depth-$\mu$P for Adam in Appendix K.

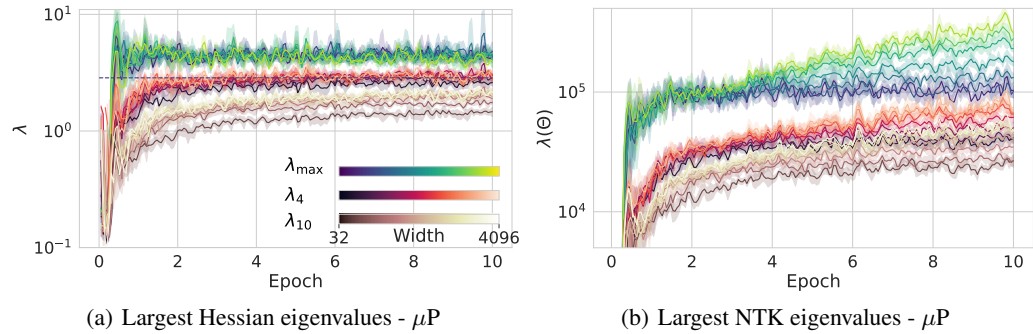

(a) Largest Hessian eigenvalues - $\mu$P          (b) Largest NTK eigenvalues - $\mu$P

Figure 2: (a) The top Hessian eigenvalues exhibit a progressive increase to a threshold, with larger eigenvalues showing precise Super Consistency, while lower eigenvalues show finite-size accumulation at small width in the initial phase of training. (b) Top eigenvalues of the NTK matrix $\Theta$. As opposed to the top eigenvalues of the Hessian, these exhibit evident finite-size accumulation during training. Model: 3-layer ConvNet, $\tau = 0$, $\eta_0 = 0.7$ (optimal). Details in Sec. J.

## 3   Super Consistency of the Optimization Landscape

In this work, we analyze the landscape through the lens of the preconditioned Hessian $\gamma^2 H_t$, where $H_t := \nabla_\theta^2 \mathcal{L}(\theta_t) := \sum_\mu \nabla_\theta^2 \mathcal{L}(f_t(x_\mu)) \in \mathbb{R}^{P \times P}$, as $\theta_t$ evolves with gradient descent. The Hessian is a key object in optimization theory [21], information geometry [22, 23], and deep learning theory [17, 24, 16, 13] and its relation to optimal step sizes is often used to design second-order optimizers [23, 25–27]. In Figure 1, we can observe that learning rate transfer correlates with strong alignment across model sizes of the Hessian top eigenvalue dynamics, a property which we name *Super Consistency*. The choice of the preconditioning factor $\gamma^2$ ensures the right scaling with respect to the width $N$, as the theory will justify. We also provide an intuition and an extension to Adam in Appendix J.1. Unless stated otherwise, every experiment is conducted with the this preconditioning factor $\gamma^2$ set according to the corresponding parametrization.

More concretely, Super Consistency refers to when certain aspects of the loss landscape and of the predictor $S_N(t)$ (in this paper $S_N(t)$ refers to the NTK's and loss Hessian's eigenvalues or the loss itself) exhibit the following two properties:

- At realistically large $N$, $S_N(t)$ does not deviate significantly from its limit $S_\infty(t) := \lim_{N \to \infty} S_N(t)$. This is what is referred to as consistency in Vyas et al. [9].

- $S_N(t)$ does not accumulate significant finite-size effects over time, i.e. the curves of $S_N(t)$ and $S_\infty(t)$ remain close over a sustained period of training.

With respect to the experiment illustrated in Fig. 1, notice that the curves of the loss (center) at different widths show progressive and significant deviations, thus violating Super Consistency. On

the other hand, the sharpness dynamics for $\mu$P qualitatively exhibit little-to-no deviations. Also, notice that we assume existence of the limit $\lim_{N \to \infty} S_N(t)$. For those parametrizations (e.g. *standard parametrization*[6]) that do not have a well-defined limit, $S_N(t)$ diverges at large $N$ and Super Consistency is trivially violated.

We now turn to the analysis of the Hessian spectrum and observe the following:

***Observation***: *in $\mu$P (and Depth-$\mu$P), Hessian eigenvalues are super consistent along the optimisation trajectory. Smaller eigenvalues have progressively different dynamics.*

In Fig. 2 (a), we train a residual network on CIFAR-10 (a 10 classes image classification task) using cross-entropy loss, and show super consistent dynamics of three of the top $k = 10$ eigenvalues. The choice of $k = 10$ is guided by Gur-Ari et al. [17], where it is observed that stochastic gradient descent happens in a small subspace where the gradient lies in the space spanned by top $k$ Hessian eigenvectors (where $k$ is the number of classes). Thus, our results show that the curvature along the training trajectory is preserved super consistently at different scales, thus suggesting that the geometry of the landscape across the trajectory is preserved across model size. Lower order eigenvalues tend to accumulate finite-size effects in the first phase of training, and stabilize at lower thresholds for smaller width models. We discuss the effect of lower order eigenvalues through the Hessian trace in Appendix G. Finally, to make sure that Super Consistency holds along the training trajectory regardless of the tiny-subspace assumption, in Appendix I we track the *directional sharpness*, that measures the curvature along the gradient direction.

To give a quantitative measure to the finite-size accumulation property, we measure deviations over time by estimating the following quantity:

$$g(t) := |S_N(t) - S_\infty(t)|. \tag{3}$$

When $g(t)$ increases over time (up to fluctuations), Super Consistency is violated. We illustrate this in Fig. 3 (b, c), where we compute the left hand side of Eq. 3 for the loss $\mathcal{L}(f_t)$ and the NTK's largest eigenvalue $\lambda_{max}(\Theta)$. To estimate the infinite width limit, we use a very large-width model as a proxy. Notice how under $\mu$P the the loss dynamics progressively diverge from the infinite width model, indicating a finite-size accumulation over time. The same holds for $\lambda_{max}(\Theta)$. To study the rate of divergence $g(t)$, we fit a power law of the form $y = at^\beta$ to the observations. A larger $\beta$ indicates a higher divergence rate. Notice how $\beta > 0.6$ for the loss, and $\beta \approx 2$ for $\lambda_{max}(\Theta)$, indicating quadratic divergence. In comparison, in Fig. 3 (left), we show Super Consistency of the sharpness, in that finite-size effects are not accumulated over time (10 epochs). Finally, both in Fig. 2 and 3 notice how the sharpness is not just *converging* in width, but in fact *width-independent*. This might be due to the fact that the threshold is a stable attractor of the dynamics [28].

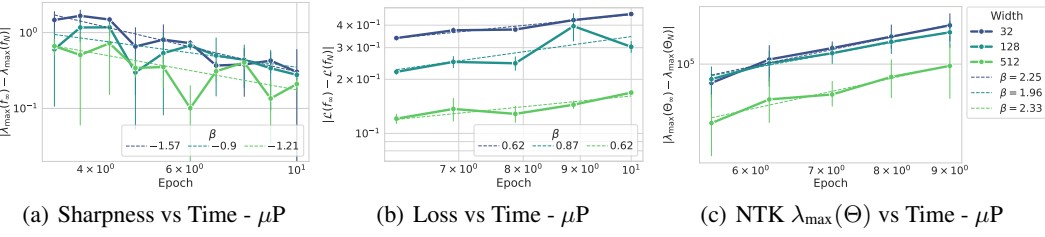

|                          |                          |                          |
|:------------------------:|:------------------------:|:------------------------:|
| (a) Sharpness vs Time - $\mu$P | (b) Loss vs Time - $\mu$P | (c) NTK $\lambda_{max}(\Theta)$ vs Time - $\mu$P |

Figure 3: (a) Convergence rate of the sharpness at finite width $N$ to the infinite limit proxy. Note that the distance approaches $0$ as the training time increases. (b) Convergence rate of the loss at finite width $N$ to the infinite limit proxy. Note that the loss accumulates finite-size effects over time and the distance to the proxy increases. (c) Convergence rate of the top NTK eigenvalues over time to the infinite limit proxy. Similar to the loss, this also accumulates finite-size effects over time. Details: infinite limit proxy is width $4096$, model is ConvNet, $\tau = 0$, $\eta_0 = 0.7$.

## 4 Super Consistency and Learning Rate Transfer

**Sharpness and Edge of Stability.** We now focus on the *sharpness* $\lambda := \lambda_{max}(\gamma^2 H)$, defined as the largest eigenvalue of the Hessian. In the theory of smooth convex [21], nonconvex [29], and stochastic [30] optimization, the sharpness plays a crucial role in in the guarantees of convergence

of gradient methods and selection of the optimal step size. For instance, for a quadratic objective, $\lambda_t = \lambda$ is constant and gradient descent would diverge if the learning rate satisfies $\eta_0 > 2/\lambda$, and training speed is maximized for $\eta_0 = 1/\lambda$ (LeCun et al. [2], page 28). Beyond this classical example, which assumes constant Hessian, the descent lemma [21] states that $\mathcal{L}(\theta_{t+1}) \leq \mathcal{L}(\theta_t)$ if $\eta \leq \frac{2}{\beta}$ where $\beta := \sup_\theta \|\nabla^2 L(\theta)\|_2$, and $\|\nabla^2 L(\theta)\|_2$ is the sharpness at $\theta$. When it comes to deep neural networks, $\lambda_t$ is generally observed to increase during training (*progressive sharpening*): in the early phase of training it increases [31, 32] and then it decreases close to convergence [10]. Under full batch gradient descent training, the sharpness consistently rises above the *EoS* threshold of $2/\eta_0$ [16].

**Conditions for hyperparameter transfer.** The empirical success of hyperparameter transfer crucially relies on the following two observations, constituting a "theoretical puzzle" [8].

1. The optimal learning rate is preserved across widths/depths, indicating very fast convergence with respect to the scaling quantity.

2. The models show consistent improvement in training speed with respect to the scaling quantity (i.e. there is a clear "wider/deeper is better" effect), indicating that the loss dynamics have not yet converged to the limiting behaviour predicted by the theory.

In this section, we study the role of Super Consistency in learning rate transfer. We focus on the dynamics of sharpness $\lambda_{\max}$ across training, due to its well-established connection to optimization theory and step size selection, as well as better computational tractability than the full Hessian spectrum. We provide extensive studies of other relevant spectral quantities (i.e. Hessian and NTK eigenvalues) in Appendix G.

***Observation****: in μP (and Depth-μP), the sharpness $\lambda$ is super consistent along the training trajectory, while for NTP the sharpness decreases in width. This correlates with presence/absence of hyperparameter transfer.*

In Fig. 1 we train a two-layer convolutional network under the μP and NTP scalings with cross entropy loss, while keeping track of the sharpness at fixed gradient step intervals. The top row shows the dynamics of $\lambda$. Notice how the sharpness' behaviour is qualitatively different in the two parameterizations: in μP it reaches a width-independent value which is close to the EoS threshold of $2/\eta_0$. On the other hand, in NTP we observe a progressive diminishing of the sharpness with width, as previously observed for Mean-Square-Error loss by Cohen et al. [16].

We then study the effect of depth under the Depth-μP model of Eq. 1. In Fig. 4 (left), we show that the sharpness' dynamics are also super consistent across depth, although progressively diminishing from the EoS threshold. This suggests that EoS is not necessary for the learning rate to transfer, but the consistency of the sharpness dynamics is.

**Other feature learning parameterizations.** Finally, we study the effect of other feature parameterizations that *do not* exhibit learning rate transfer. In particular, we study the Depth-μP scaling of the residual branches in residual networks with multiple layers per residual branch - denoted by $k$ (i.e. each branch has multiple weight matrices and non linearities). A typical example is the Transformer architecture, which has multiple layers per block in both the attention and fully connected blocks. This parameterization, although it learns features in the infinite depth limit, it is *lazy within each residual branch* [8, 7]. The results are in Fig. 4. Notice how the sharpness dynamics are not super consistent, in that they accumulate finite-size effects over time. We study other parametrizations, including those without a stable limit in Appendix B, showing compatible results with those presented here. The observation that sharpness dynamics exhibit greater consistency compared to loss dynamics suggests that under μP scaling, although models with larger capacity fit the data faster, the paths taken by various models through the optimization landscape show a surprisingly uniform curvature.

## 4.1 Feature Learning and Progressive Sharpening

We now study the effect of feature learning in the sharpness dynamics. Following the Gauss-Newton decomposition [25, 12], the Hessian can be decomposed as a sum of two matrices $H = \mathcal{G} + \mathcal{R}$, where

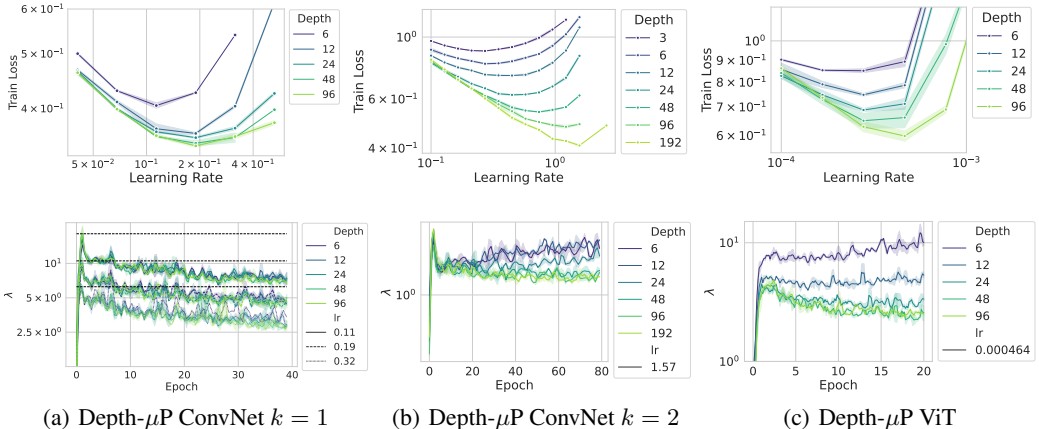

| (a) Depth-$\mu$P ConvNet $k = 1$ | (b) Depth-$\mu$P ConvNet $k = 2$ | (c) Depth-$\mu$P ViT |
|---|---|---|

Figure 4: Depth-$\mu$P extensions with top row showing transfer plots and bottom row the sharpness evolution. (a) ConvNets with 1 layer per block exhibit both hyperparameter transfer and sharpness Super Consistency. (b) ConvNets with 2 layers per block. The model has a lazy behavior within each block, and no transfer. The sharpness starts accumulating finite-size effects during training, violating Super Consistency. (c) ViTs also have $k > 2$ blocks per layer by design, and thus have a similar behaviour. Details: (a), (b) are trained with SGD, with widths 128 and 32 respectively; (c) is trained with Adam, with the learning rate scaled by $1/\sqrt{L}$ [8]. See Fig. 22 for convergence rates.

$\mathcal{G}$ is the Gauss-Newton (GN) matrix and $\mathcal{R}$ depends on the Hessian of the model. For MSE loss,

$$\mathcal{G} = \sum_{i=1}^{|\mathcal{D}|} \nabla_\theta f(x_i) \nabla_\theta f(x_i)^\top = K^\top K \quad \text{and} \quad \mathcal{R} = \sum_{i=1}^{|\mathcal{D}|} \nabla_\theta^2 f(x_i)(y_i - f(x_i)),$$

where $K \in \mathbb{R}^{|\mathcal{D}| \times P}$ is a matrix where each row is $\nabla_\theta f(x_i)$ (i.e. the Jacobian of $f(x_i)$), and $y_i \in \mathbb{R}$ is the label. One can readily see that the NTK matrix can be written as $\Theta(f_\theta) = KK^\top$, thus the NTK and $\mathcal{G}$ share the same nonzero eigenvalues. In Figure 5, we show that under $\mu$P the sharpness evolution is dominated by the $\mathcal{G}$ matrix consistently across different widths, while for NTP the sharpness evolution slows down when increasing the width. Since in the limit the NTK matrix is fixed for NTP, while it evolves with time for $\mu$P, these results provide further supporting evidence for the role of feature learning in the evolution of the hessian. While this argument strictly holds for MSE loss, it can be generalized to any twice differentiable loss function, albeit with some caveats. In Appendix D, we generalize the setting, analyze the cross-entropy loss and perform validating experiments, confirming the conclusions drawn here. Finally, in Appendix H, we show that our results remains valid in a random feature model, where the NTK matrix is fixed at initialization at any finite width. In Section 5 we revisit the above claims more precisely in a simplified setting, providing further intuition on the sharpness dynamics and learning rate transfer.

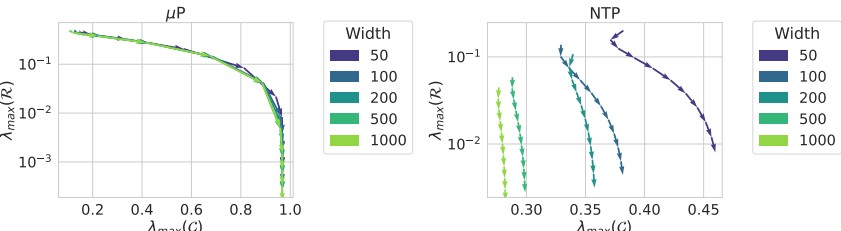

Figure 5: Evolution of the top eigenvalues of the Hessian components $\mathcal{G}$ and $\mathcal{R}$ for a two-layer linear network trained on random data under MSE loss. The vector field characterizes the evolution during training for a fixed learning rate. Top: $\mu$P. Note how $\mathcal{G}$ drives the initial change super consistently. Bottom: NTP. For wider networks the sharpening phase reduces, since the network is approaching the limit where the NTK is fixed to its value at initialization.

**Large scale experiments.** In App. F, we perform more experiments to validate the connection between the consistency of the sharpness' dynamics and learning rate transfers across datasets (Tiny-

ImageNet, Wikitext), architectures (ViT, GPT-2 [33]), and optimizers (Adam [34] and AdamW [35]). We find these results to be consistent with those in the main text.

**End of training dynamics.** In App. E.1 (Fig. 12), we study the width dependence of the sharpness at the late phase of training. It is well-known that for cross-entropy loss, a phase transition happens where the sharpness starts to decrease [16]. We found that even for $\mu P$ this transition point is width-dependent, with a consequent slight shift in optimal learning rates during this late phase. Again, these results are in line with our results that super consistent sharpness facilitates transfer.

**Batch size ablation.** We repeat the experiment in Fig. 1 with increasing batch size, observing that the threshold is reached across all the tested batch sizes, thus not affecting learning rate transfers. For larger batches, a close-to-EoS threshold is reached across more learning rates. Results are summarized in Fig. 13 and 14 in App. E.

## 5 Case study: Two-Layer Linear Network

We now revisit and validate our intuition and empirical findings in Sec. 4 through the lens of a two-layer neural network with linear activations and $L2$ loss. Our purpose is to characterize the dynamics of $\mu P$ and NTP at the edge of stability through the lens of a simple example that shares a similar phenomenology with the more complex scenarios observed in the last section (see Fig. 10, App. C). In particular, the theory justifies the preconditioned Hessian $\gamma^2 H \propto NH$ as the right object of study when it comes to the sharpness computations (see Prop. 5.3). Also, it provides an intuition to the width-independent evolution of the sharpness. Our setting is similar to the one leading to the insightful analysis of *EoS* in [28, 36]. Compared to these works, we do not limit the analysis to a single datapoint or to vanishing targets [1].

**Notation and assumptions.** Consider a dataset of $|\mathcal{D}|$ datapoints in $D$ dimensions $X \in \mathbb{R}^{|\mathcal{D}| \times D}$ ($|\mathcal{D}| \geq D$), and labels generated through a latent ground-truth vector $w_* \in \mathbb{R}^D$, that is $Y = Xw_*$. The neural network we use here is parametrized by weights $W^0 \in \mathbb{R}^{D \times N}$, $W^1 \in \mathbb{R}^{N \times 1}$, where $N$ is the width. To simplify the notation in our setting, we name $E := W^0$ and $V := W^1$: $f(X) = \frac{1}{\gamma\sqrt{ND}} XEV$, $\mathcal{L}(E, V) = \frac{1}{2}\|f(X) - Y\|^2$. We initialize each entry of $E, V$ i.i.d. Gaussian with mean zero and variance 1. Recall that $\gamma_{\text{NTP}} = 1$, $\gamma_{\mu P} = \sqrt{N}$. We train with gradient descent (GD) with a learning rate $\eta = \eta_0 \gamma^2$. Empirically, we observed (Fig. 10, App. C) that picking $|\mathcal{D}| = D$ and data $X = I_D$ ($I_D$ is the $D \times D$ identity matrix) is sufficient to track most of the crucial features of $\mu P$/ NTP explored in this paper, except the "wider is better" effect which here is less apparent due to the simple hypothesis class. The loss function reduces to:

$$\mathcal{L}(E, V) = \frac{1}{2}\|w - w_*\|^2, \quad \text{with} \quad w := \frac{1}{\gamma\sqrt{ND}} EV. \tag{4}$$

Finally, we reparametrize the model by defining:

$$e := \frac{1}{ND} EE^\top \in \mathbb{R}^{D \times D}, \quad v := \frac{1}{ND} V^\top V \in \mathbb{R}_{\geq 0}. \tag{5}$$

Note that using a learning rate $\eta_0 \gamma^2$ when optimizing $\mathcal{L}$ is equivalent to using a learning rate $\eta_0$ when optimizing $\gamma^2 \mathcal{L}$. Next, we characterize how $e, v$ evolve through time, and give conclusions for $\mu P$.

### 5.1 Dynamics and Edge of Stability in Latent Space

We now show that at any value of the width $N$, under GD on the original network parameters $(E, V)$, the dynamics of $w, e, v$, can be described completely through a self-contained dynamical system in $(1 + D + D^2)$ dimensions. This property is surprising because the original dynamical system described by GD on the variables $E, V$ lives in $N(D + 1)$ dimensions. Concretely, this means we can study the Hessian dynamics at different network widths in the same space.

**Theorem 5.1** (Evolution Laws). *Let $(E, V)$ evolve with GD at stepsize $\eta = \eta_0 \gamma^2$ on the loss of Eq. 4. The evolution of $(w, e, v)$ is completely described by the following self-contained equation:*

---

[1]If our dataset has cardinality 1, then the NTK is a scalar. If targets vanish, for 2-layer linear networks with $L2$ loss, NTP and $\mu P$ induce the same loss on the parameters ($\gamma$ cancels).

*let the $^+$ denote updated quantities,*

$$w^+ = w - \eta_0(v \cdot I_D + e)(w - w_*) + \frac{\eta_0^2 \gamma^2}{ND}(ww^\top - w_*w^\top)(w - w_*).$$

$$e^+ = e + \frac{\eta_0 \gamma^2}{ND}\left[-2ww^\top + w_*w^\top + ww_*^\top\right] + \frac{\eta_0^2 \gamma^2}{ND}\left[vww^\top - vw_*w^\top - vww_*^\top + vw_*w_*^\top\right].$$

$$v^+ = v + \frac{\eta_0 \gamma^2}{ND}\left[-2w^\top w + 2w_*^\top w\right] + \frac{\eta_0^2 \gamma^2}{ND}\left[w^\top ew - 2w_*^\top ew + w_*^\top ew_*\right].$$

While the system above describes the evolution laws $(w_k, e_k, v_k) \to (w_{k+1}, e_{k+1}, v_{k+1})$, the dynamics are influenced also by initialization. In Prop. C.1 in the Appendix, we show that the only dependence in width in the evolutions laws are in the initial conditions.

Last, by analyzing the stability of the dynamical system in Theorem 5.1, we can characterize the edge of stability using tools from dynamical systems [37]. First of all, we need the following Lemma, which implies that at the minimizer ($w = w_*$), the Hessian has the same non-zero eigenvalues as the NTK $\Theta$, which only depends on $e$ and $v$.

**Lemma 5.2** (GN bound). *Let $\gamma^2 \nabla^2 \mathcal{L} = \mathcal{G} + \mathcal{R}$ be Gauss-Newton decomposition[2] (see Sec. 4.1) of the Hessian for the loss in Eq. 4, with $\mathcal{G} = K^\top K$, where $K \in \mathbb{R}^{D \times (ND+N)}$.*
*Let us denote the NTK matrix $\Theta = KK^\top \in \mathbb{R}^{D \times D}$. Then*

$$\Theta(E, V) = e + v \cdot I_D$$

*and*

$$|\lambda_{\max}[\gamma^2 \nabla^2 \mathcal{L}(E, V)] - \lambda_{\max}[\Theta(E, V)]| \leq \sqrt{\frac{\gamma^2}{ND}}\|w - w_*\|_2.$$

We stress that this result implies that evolution of the NTK (i.e. *feature learning*) goes hand in hand with the evolution of the sharpness, as we empirically show in Sec. 4.1. We are now ready to state the result on the sharpness at convergence.

**Proposition 5.3** (EoS). *Let $(E, V)$ evolve with GD with stepsize $\eta = \eta_0 \gamma^2$ on the loss of Eq. 4 towards a minimizer $(E_*, V_*)$. Assume the corresponding solution in latent space $(w_*, e_*, v_*)$ is marginally stable [3]. Then, $\lambda_{\max}[\gamma^2 \nabla^2 \mathcal{L}(E_*, V_*)] \in \left[\frac{2}{\eta_0}, \frac{2}{\eta_0} + \frac{\eta_0 \gamma^2 \|w_*\|^2}{ND}\right]$.*

**Implications for NTP.** Consider $\gamma = 1$ in Thm. 5.1. The dynamics of $(w, e, v)$ are *width-dependent*. Let us take $N \to \infty$ in the equation above to amplify this effect: the system becomes *linear*

$$w^+ = w - \eta_0(v \cdot I_D + e)(w - w_*), \ \ e^+ = e, \ \ v^+ = v.$$

While $w$ evolves from $w_0$ as expected from standard NTK theory [4], $e, v$ stay clamped at initialization. Applying Lemma 5.2 with $\gamma = \mathcal{O}(1)$, the Hessian and the NTK have the same largest eigenvalue at large width (at rate $O(\sqrt{N})$). This agrees with our intuition, as under NTP the predictor converges to a linear model in the large $N$ limit, and thus $\mathcal{R}$ vanishes. Also, this proves that the sharpness has no dependency on the learning rate in the width limit (we observe this, e.g., in Fig. 1 and throughout all our experiments). This derivation is also in line with our discussion in Sec. 4.1: we only have sharpening under feature learning, and for the same reason we cannot observe NTP at the edge of stability as $N \to \infty$ (see stepsize dependency in Prop. 5.3), as also noted empirically by [16].

**Implications for $\mu$P.** The following result immediately follows by inspection of the equations in Thm 5.1, combined with Prop. C.1.

**Corollary 5.4.** *Consider $\mu P$ ($\gamma = \sqrt{N}$) and let $(E, V)$ evolve with GD with stepsize $\eta = \eta_0 \gamma^2$ on the loss of Eq. 4. Then, the equations governing the evolution of $(w, e, v)$ (defined in Thm. 5.1) in latent space have **no width dependency** – this holds at any finite width and not just at the limit. Initialization of $(w, e, v)$ is instead width-dependent, yet the error from $N \to \infty$ case scales in expectation like $1/\sqrt{N}$.*

---

[2] Recall: $|\mathcal{D}| = D$ in our simplified setting.
[3] In a dynamical system sense: some eigenvalues of the Jacobian have unit norm, others have norm $< 1$.

The corollary shows that $\mu$P trajectories at different widths align in the latent space $(w, e, v)$, albeit with a vanishing perturbation in the initial condition (see Prop. C.1). While NTP's dynamics for $e$ and $v$ become slower as the width increases, for $\mu$P their evolution laws are width independent. This implies that if the dynamics converge towards a minimizer for $e$ and $v$, this will be at the sharpness value predicted by Prop. 5.3. Under $\mu$P, where $\gamma^2 \propto N$, this value will be *width-independent*, as Super Consistency would suggest. We stress that Prop. 5.3 characterizes the sharpness at convergence (i.e. at infinite time). At finite time, there is still a discrepancy between the $\lambda_{\max}(\Theta)$ and the sharpness of the order of the residual term $1/\sqrt{D}\|w - w^*\|$ (Lemma 5.2). Finally, we stress that Prop. 5.3 prescribes the right scaling for the Hessian by including the preconditioning factor of $\gamma^2$. Thus, we do not prove that at any finite time, the whole sharpness trajectory is width-independent, nor we are estimating converge rates in $N$ at finite time. Indeed, there will be a finite-size dependence coming from the initial conditions. We leave a precise characterization of the whole sharpness dynamics across the training trajectory for future work.

## 6 Discussion & Conclusions

**On Feature Learning Parametrizations.** In this paper, we have shown how certain properties of the loss Hessian evolve almost identically across training for different model sizes, and named this property *Super Consistency*. We have also compared the sharpness dynamics under different scaling limits and parameterizations, and related Super Consistency of the landscape to learning rate transfer. Beyond being able to distinguish feature learning (rich) and kernel (lazy) parametrization, we have also shown how other suboptimal feature learning parametrizations have sharpness dynamics violating Super Consistency through finite-size accumulations. This seems to suggest that Super Consistency of the landscape is an important discriminant when it comes to hyperparameter transfer beyond the rich/lazy regimes. We foresee that our paper could spark further research interest at the intersection between the scaling limits of neural networks and optimization theory.

**On the NTK and Hessian dynamics.** In Section 4.1 we have drawn the connection between progressive sharpening and NTK evolution in the early phase of training. However, Figure 3 (b), shows how the NTK eigenvalues at different widths accumulate finite-size effects over time and diverge from each other, while the Hessian eigenvalues are Super Consistent. This suggests that other forces are at play after progressive sharpening, such as *Self-Stabilization* [38]. In fact, progressive sharpening on one hand, and Self-Stabilization on the other, make the stability threshold a stable attractor of the sharpness dynamics. Gaining theoretical understanding for these complex interactions in the context of scaling limits is an exciting area of future research.

**Design of Step-size Tuners.** In most of our experiments, we rely on a constant step size $\eta_0$. However, an alternative is to use a step-size tuner, i.e. to automatically choose $\eta_0$ based on some criteria of the local landscape [39]. Our results open directions into some possible investigations and design choices for new step size tuners. For instance, do step size tuners transfer with the width and depth of the architecture? Given our results on the role of warmup schedule to improve transfer, it seems plausible to design step size tuners that use EoS results to achieve optimal learning rate transfer under different parameterizations.

**Limitations.** One of the underlying assumptions of the argument presented here is that the sharpness is an important property of the landscape when it comes to step size selection. Indeed, the results in Cohen et al. [16] establish a more intricate relationship between sharpness and learning rate. We discuss this in Sec. A.2. Overall, our theory on Super Consistency does not exclude the existence of other factors that might influence the optimal learning rate. Hyperparameter transfer requires Super Consistency of the landscape, thus we expect other potential factors to have this property. Finally, we note that due to the high computational cost of Hessian estimation, we do not perform experiments at a larger scale than presented here. It would be interesting to see if Super Consistency still holds at an even larger scale.

## Acknowledgements

The authors would like to thank Bobby He, Imanol Schlag, Dayal Kalra, Tiago Pimentel and Gregor Bachmann for providing insightful feedback on early versions of this manuscript. LN would also like to thank Blake Bordelon, Boris Hanin and Mufan Li for the stimulating discussions on the topic

of scaling limits that helped inspiring this work. AO acknowledges the financial support of the Hector Foundation. LN acknowledges the support of a Google PhD fellowship. AM acknowledges the support of a Kempner Fellowship.

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

# Appendix

## Table of Contents

# A Related works

**Hyperparameter search.** Hyperparameter tuning [40] has been paramount in order to obtain good performance when training deep learning models. With the emergence of large language models, finding the optimal hyperparameters has become unfeasible in terms of computational resources. Classical approaches based on grid searching [41] over a range of learning rates, even with improvements such as successive halving [42, 43] require training times on the order of weeks on large datasets, making them largely impractical. Bayesian optimization [44, 45] methods aim to reduce the search space for the optimal HPs by choosing what parameters to tune over in the next iteration, based on the previous iterations. A different approach for HP search involves formulating the problem as an optimization over the HP space and solving it through gradient descent [46–48].

**Learning rate transfer.** While meta-learning and neural architecture search (NAS) literature provide methods for finding the optimal learning rate in neural networks, these are still dependent on the model size and can become costly for larger architectures. Parameter transfer methods have been studied in the literature, for learning rate transfer across datasets [49], and in the context of reusing previous hyperparameter optimizations for new tasks [50]. Perrone et al. [51] and Horváth et al. [52] proposed methods based on Bayesian optimization for hyperparameter transfer in various regimes. Yang and Hu [5], Yang et al. [6, 8], and Yang and Littwin [53] used the tensor programs framework to derive a model parameterization technique which leads to feature learning and learning rate transfer through width and depth, termed $\mu$P. Bordelon et al. [7] used the DMFT framework [20, 54] to derive the Depth-$\mu$P limit, leading to optimal learning rate transfer across depth is models with residual connections. For MLPs, Jelassi et al. [55] the depth dependence of the $\mu$P learning rates. Finally, conditions on the networks and its dynamics to achieve hyperparameter transfer have been analyzed in Yaida [56].

**Training on the Edge of Stability.** The choice of learning rate has been coined as one of the important aspects of training deep neural networks [57]. One phenomenon studied in the optimization literature is the fact that under gradient descent (GD), neural networks have sharpness close to 2/step size, termed the Edge of Stability (EoS) [16, 38, 58–60], with the extension to Adam being introduced by Cohen et al. [61]. Iyer et al. [62] study the maximal learning rate for ReLU networks and establish a relationship between this learning rate and the width and depth of the model. Smith and Topin [63] and Y. Li et al. [64] study the effect of large initial learning rates on neural network training. Lewkowycz et al. [15], Kalra and Barkeshli [65], and Kalra et al. [28] show empirically that the learning rate at initialization can lead the "catapult" phenomena. Song and Yun [36] analyze the trajectory of gradient descent in a two-layer fully connected network, showing that the initialization has a crucial role in controlling the evolution of the optimization. Finally, early evidence of Super Consistency was shown in Fig. 3 of Sagun et al. [11], where it was shown that at end of training the Hessian spectrum is similar across model sizes.

**Scaling limits** The study of scaling limits for neural network was pioneered by the seminal work of Neal [66] on the equivalence between infinite width networks and Gaussian Processes, and more recently extended in different settings and architectures [67–69] and under gradient descent training, leading to the Neural Tangent Kernel (NTK) [4, 19, 18, 70] or "lazy" limit [71]. The rich feature learning infinite width limit has been studied using different frameworks, either Tensor programs [5] or DMFT [20, 54]. The main motivation behind these works is to maximize feature learning as the width is scaled up. In the two-layer case, the network's infinite-width dynamics have also been studied using other tools, such as optimal transport [72, 73] or mean-field theory [74]. The infinite depth analysis of $1/\sqrt{\text{depth}}$-scaled residual networks was introduced in [75], and later applied to the Transformers (used here) in [76]. The infinite width-and-depth limit of this class of residual networks have been studied in Hayou and Yang [77] and Hayou [78] at initialization and in Bordelon et al. [7], Yang et al. [8] for the training dynamics. Without the $1/\sqrt{\text{depth}}$-scaling, the joint limit has mainly been studied at initialization [79–84]. Deviations from the infinite dynamics can also be studied using perturbative approaches [85–87]. Finally, it is worth mentioning that a method to control and measure feature learning have been recently proposed in Chizat and Netrapalli [88].

**Scaling limits and Hyperparameter transfer** It is worth mentioning that while for width limits feature learning is a clear discriminant between NTK and mean-field limits, the issue becomes more subtle with depth limits. In fact, there exist a family of infinite depth limits that admit feature

learning ($\alpha \in [1/2, 1]$ in Eq. 1, with an appropriate depth correction to the learning rate). [8] classifies the depth limits in terms of the feature diversity exponent, a measure that quantifies the diversity between the features of different layers. With respect to this measure, $\alpha = 1/2$ is the one that maximizes it. Bordelon et al. [7] try to quantify the finite-size approximation to the infinite (continuous) model, arguing that hyperparameter transfer is achieved faster for models that have lower discretization error to the infinite model's dynamics.

## A.1 Learning Rate Transfer and Scaling Limits

Our results on the early training dynamics complement the analysis of Vyas et al. [9] and Kalra et al. [28] on the consistency of the loss and the sharpness in the first few steps of training. However, we further extend it, showing how while the loss curves depart later in training (with "wider/deeper being better"), the consistency of the sharpness' dynamics is maintained longer in time and does not accumulate finite-size effects over time. Furthermore, our work explains some of the experimental results on the lack of progressive sharpening under the NTP parameterization [16] (Appendix H), by relating it to the lack of feature learning and the consequent absence of hyperparameter transfer. Our results on the role of feature learning are also compatible with [89], where it is theoretically shown that the non linear dynamics exhibits progressive sharpening in a simple non linear model. Our results are also in line with the recent experiments on the evolution of the sharpness for ReLU MLPs trained with gradient descent under $\mu$P [28] (e.g. Figure 1). We extend these results to include the width (in)-dependence behaviour of the convergent stability point, a crucial aspect for successful hyperparameter transfer.

Finally, It is worth mentioning that there exist a family of infinite depth limits that admit feature learning ($\alpha \in [1/2, 1]$ in Eq. 1, with an appropriate depth correction to the learning rate). Most of our experiments focus on models with a single layer per residual block, which exhibit transfer more consistently under the $\mu$P- $\sqrt{\text{depth}}$ setting (i.e. $\alpha = 1/2$) and it is considered optimal in terms of feature diversity across blocks [8]. We adopt the same depth scaling with our experiments with Transformers — which have multiple layers per block. Under this setting, both Bordelon et al. [7] (e.g. Fig. 3) and Yang et al. [8] (Fig. 16) show good (albeit at time slightly worse) transfer across depth with Adam. More broadly, we expect that a scaling limit that admits learning rate transfer would find a corresponding width/depth-independent behaviour in the sharpness dynamics.

## A.2 Sharpness and Optimal Step Size

One of the underlying assumptions to explain why super consistent sharpness causes hyperparameter transfer is that the sharpness influences the optimal learning rate. Although this is well-established in classic optimization [2], Edge of Stability tells a story of a more intricate relationship: the choice of the learning rate also influences the sharpness. Also recent works on step size tuners show evidence of a more complex interaction in the joint dynamics of step size and sharpness [39], a relation that still has to be fully understood and could be leveraged to design better step size tuners. However, the sharpness is still arguably a very good proxy for understanding the trainability of the model at large learning rates. For instance, Gilmer et al. [90] argue that maintaining a small sharpness in neural network optimization favors training at large learning rates. Interestingly, in Cohen et al. [16] (Appendix F) it is shown that choosing the step size as $1/\lambda_t$ at any step is suboptimal. The reason could be that gradient directions are not aligned with the largest curvature. Alternatively, one could claim there is another, more sophisticated, functional relationship between sharpness and the maximum step size allowed. Indeed, the "catapult mechanism"[15] shows the maximum stable learning rate is $c_{arch}/\lambda_0$, where $c_{arch}$ depends on the architecture. On a related note, Gur-Ari et al. [17] shows that after the first few steps, the gradient direction is aligned with the top Hessian eigenvectors, underlying once again the importance of the sharpness and the first few eigenvalues in this phase of training, which we show here to be super consistent (Fig. 2).

# B  Experiments in the absence of a valid scaling limit

## B.1  $\mu$P *without* $1/\sqrt{\text{depth}}$ scaling of the residual branches

We first study the effect of increasing the depth in a model without the $1/\sqrt{L}$-scaling of the residual branches, which results in exploding activations (and sharpness) as the depth increases. See Figure

[6](#), where we first show that (top row) there is no learning rate transfer, and there is no consistent sharpness either. For instance, notice how the sharpness for the model of depth 24 quickly reaches its EoS value at its optimal learning rate of 0.068 (and larger models, e.g. depth 36 diverge). On the other hand, for the same learning rate the smaller depth models struggle to reach EoS at the same speed. This suggests that for the smaller-depth model, the learning rate can be safely increased. Indeed, the optimal learning rate for the smaller depth model is significantly larger, where the model reaches the EoS value very fast (optimal lr: 0.53). The larger depth models are not trainable at these larger learning rates.

In the bottom row of Figure [6](#), we show that by adding a *linear warmup* scheduler in the first phase of training, the network is progressively more trainable at larger step-sizes, and the sharpness reaches its stability threshold at any width/depth that is trainable [90]. This suggests that the diverging initial sharpness can be counteracted with initial small learning rates. Also, we notice a better transfer, indicating that a sustained period of depth-independent sharpness can help learning rate transfer. On the other hand we do not observe good transfer after the first epoch, which correlates with the fact that the sharpness is not consistent due to a blow up with depth at initialization.

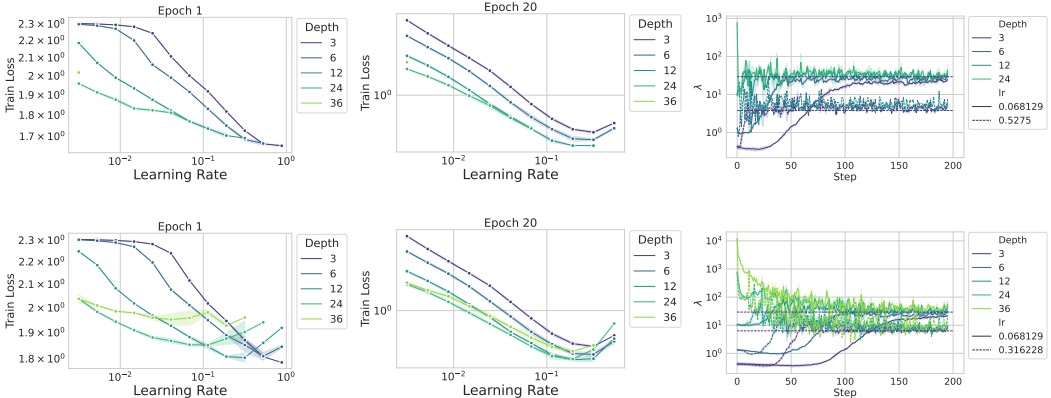

Figure 6: $\mu$P *without* $1/\sqrt{\text{depth}}$ scaling of the residual branches for ConvNets trained using SGD on CIFAR10. (Top row): no warmup. (Bottom row): 200 steps of linear warmup. Right column: sharpness dynamics in the first epoch *for the optimal learning rates for the models of depth 3 and depth 24*. Notice how in the first epoch (Left column) there is no learning rate transfer in both cases. This correlates with the fact that that the training starts at an increasing sharpness with depth, which causes no consistency the dynamics (and divergent behaviour at large learning rates). Adding warmup alleviates this issue, allowing the model to reach edge of stability and improve learning rate transfer (middle column). One step in the sharpness plot corresponds to 2 batches. Parameters: batch size= 128, epochs= 20, using data augmentation.

## B.2 Standard Parameterization (SP) experiments

Following the SP formulation introduced in [5], we analyze the sharpness and learning rate transfer under this regime in Figure [7](#). Note that our experiments use a fixed learning rate $\eta$. While Yang and Hu [5] use a width scaled learning rate in order to parameterize SP as a kernel limit in infinite width, the SP definition that we use does not have a well defined limit and thus diverges as the width is increased. For more details, we refer the reader to (Yang and Hu [5], Appendix J.3).

## B.3 $\mu$P - disabling residuals

We also provide training runs of models parameterized with $\mu$P, while disabling the residuals. Note that the models quickly become untrainable under this regime when increasing the depth. This phenomenon is due to the vanishing curvature experienced by these models, as shown in Figure [8](#), which leads to vanishing signal.

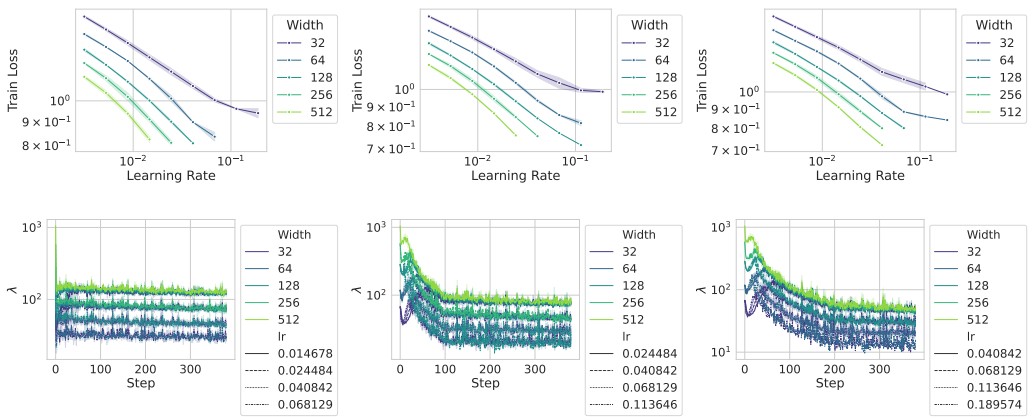

Figure 7: Standard Parameterization (SP) for ConvNets trained using SGD on CIFAR10, for varying number of learning rate warmup steps. (Left column) No warmup, (Middle column) 1000 warmup steps and (Right column) 2000 warmup steps. Note that under SP, in the beginning the training starts from a high curvature, which means that large step sizes would lead to divergent behaviour. Adding warmup alleviates this issue, allowing the model to reach edge of stability and improve learning rate transfer. One step corresponds to 10 batches. Parameters: batch size=256, epochs=20, using data augmentation.

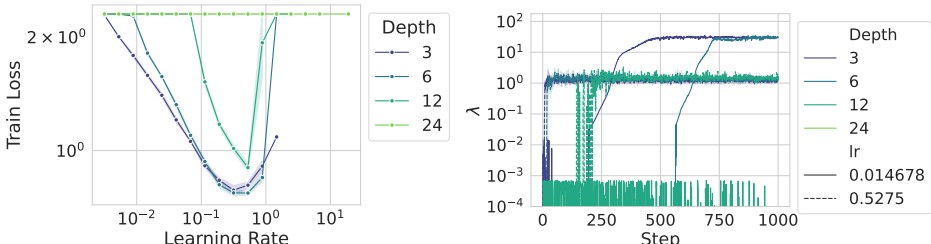

Figure 8: $\mu$P parameterization on ConvNets with residuals disabled ($\tau = 0$) trained on CIFAR10. (Left) Learning rate transfer plot, showing that under this setting, the optimal learning rate transfers, but with increasingly larger shifts when increasing the depth due to the vanishing signal. (Right) Sharpness evolution during training, showing that the dynamics are following a depth independent trajectory for the optimal learning rate (0.5275). Note that deeper models become much harder to train when residuals are disabled, motivated by the observation that the the depth 24 model suffers from vanishing curvature at larger learning rates. The spikes in the plot are due to the curvature approaching 0 in log-scale. One step corresponds to 10 batches. Parameters: batch size=256, epochs=20, using data augmentation.

## B.4 $\mu$P experiments for full batch GD

In this section, we investigate the effect of $\mu$P and on learning rate transfer and sharpness, when trained with full batch gradient descent. We subsampled 5000 sampled from CIFAR10 in a stratified fashion (i.e. 500 sampled from each of the 10 classes) and proceeded to train residual ConvNets under the $\mu$P parameterization with GD, following a similar procedure as [16]. Our findings show that under $\mu$P, when using a large enough learning rate, the models are able to achieve edge of stability, as well as have learning rate transfer. This is empirically shown in Figure 9, where we can see that while the sharpness has the typical oscillations around the EoS threshold studied by [16], it still does maintain a width-independent trend during training.

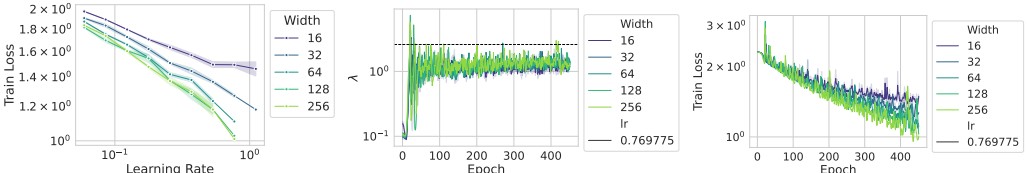

Figure 9: Residual ConvNets trained using (full batch) GD on a $5000$ sample subset of CIFAR10. (Left) Learning rate transfer plot, showing that the optimal learning rate transfers across different widths; the slight shift in the transfer plot is due to the oscillations around the EoS threshold seen in (middle) and (right). (Middle) Sharpness dynamics during training, showing a consistent width independent dynamic throughout the whole training procedure; dashed line represents $2/\eta$. (Right) Training loss dynamic during time for the optimal learning rate $(0.76)$, showing the wider-is-better behaviour of the $\mu$P parameterization, as well as the oscillations induced by the EoS regime. Parameters: batch size$= 256$, no warmup, using data augmentation.

## C  Analysis of a Two-Layer Linear Network

Recall the definition of our model:

$$L(E, V) = \frac{1}{2} \left\| \frac{1}{\gamma\sqrt{ND}} XEV - Y \right\|^2 \tag{6}$$

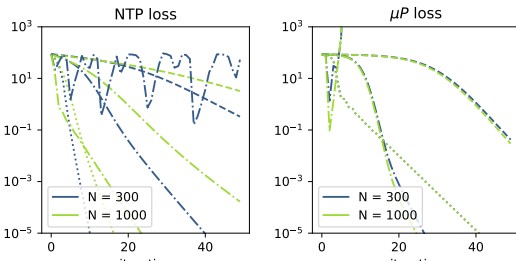

Figure 10: Evolution of loss under $\mu$P and NTP for the toy example of Section 5: $\frac{1}{2}\|\frac{1}{\sqrt{ND}\gamma}EV - w_*\|^2$, where $w_* = 1 \in \mathbb{R}^D$, $D = 100$. This is a minimal example of transfer captured by our theory: $\mu$P trajectories align. Different linestyles correspond to different values of $\eta_0$ (grid is different for $\mu$P and NTP).

Under the previous assumptions regarding the data, we have that:

$$\partial_E = \frac{1}{\gamma^2 ND} EVV^\top - \frac{1}{\gamma\sqrt{ND}} w_* V^\top.$$
$$\partial_V = \frac{1}{\gamma^2 ND} E^\top EV - \frac{1}{\gamma\sqrt{ND}} E^\top w_*.$$

### C.1  Relationship between sharpness and residual, and Gauss-Newton

The following bound leverages a Gauss-Newton decomposition and leads analytical insights supporting Sec. 4.1. Proof of Lemma 5.2, which we restate here.

**Lemma** (GN bound). *Let $\gamma^2\nabla^2\mathcal{L} = \mathcal{G} + \mathcal{R}$ be Gauss-Newton decomposition[4] (see Sec. 4.1) of the Hessian for the loss in Eq. 4, with $\mathcal{G} = K^\top K$, where $K \in \mathbb{R}^{D\times(ND+N)}$.*
*Let us denote the NTK matrix $\Theta = KK^\top \in \mathbb{R}^{D\times D}$. Then*

$$\Theta(E, V) = e + v \cdot I_D$$

---

[4]Recall: $|\mathcal{D}| = D$ in our simplified setting.

*and*

$$|\lambda_{\max}[\gamma^2 \nabla^2 \mathcal{L}(E, V)] - \lambda_{\max}[\Theta(E, V)]| \leq \sqrt{\frac{\gamma^2}{ND}} \|w - w_*\|_2.$$

The result above is actually more general: it indicates that the eigenvalues of the positive semidefinite portion of the Hessian $\mathcal{G} = K^\top K$ are fully[5] characterized by the eigenvalues a smaller matrix $\Theta = KK^\top$. Furthermore, in our model $\Theta$ has a closed form that depends only on the variables $e, v$.

*Proof.* The Hessian blocks become:

$$\partial_{EE} = \frac{1}{\gamma^2 ND} I_D \otimes VV^\top \in \mathbb{R}^{ND \times ND} \tag{7}$$

$$\partial_{EV} = \frac{1}{\gamma^2 ND} E \otimes V + \frac{1}{\gamma\sqrt{ND}} \left( \frac{1}{\gamma\sqrt{ND}} EV - w_* \right) \otimes I_N \in \mathbb{R}^{ND \times N} \tag{8}$$

$$\partial_{VE} = \frac{1}{\gamma^2 ND} E^\top \otimes V^\top + \frac{1}{\gamma\sqrt{ND}} \left( \frac{1}{\gamma\sqrt{ND}} V^\top E^\top - w_*^\top \right) \otimes I_N \in \mathbb{R}^{N \times ND} \tag{9}$$

$$\partial_{VV} = \frac{1}{\gamma^2 ND} E^\top E \in \mathbb{R}^{N \times N} \tag{10}$$

Using these definitions, we can separate the Hessian $H$ into a sum of 2 matrices, where one depends on the residual and one does not.

$$\nabla^2 \mathcal{L}(E, V) = \underbrace{\frac{1}{\gamma^2 ND} \begin{pmatrix} I_D \otimes VV^\top & E \otimes V \\ E^\top \otimes V^\top & E^\top E \end{pmatrix}}_{\mathcal{G}(E,V)} + \underbrace{\frac{1}{\gamma\sqrt{ND}} \begin{pmatrix} 0 & I_N \otimes (w - w_*) \\ I_N \otimes (w - w_*)^\top & 0 \end{pmatrix}}_{\mathcal{R}(E,V)} \tag{11}$$

hence our quantity of interest:

$$\gamma^2 \nabla^2 \mathcal{L}(E, V) = \underbrace{\frac{1}{ND} \begin{pmatrix} I_D \otimes VV^\top & E \otimes V \\ E^\top \otimes V^\top & E^\top E \end{pmatrix}}_{\mathcal{G}(E,V)} + \underbrace{\sqrt{\frac{\gamma^2}{ND}} \begin{pmatrix} 0 & I_N \otimes (w - w_*) \\ I_N \otimes (w - w_*)^\top & 0 \end{pmatrix}}_{\mathcal{R}(E,V)}, \tag{12}$$

where $w = \frac{1}{\gamma\sqrt{ND}} EV$.

**Study of $\mathcal{G}$.** Note that

$$\mathcal{G}(E, V) = K^\top K, \qquad K^\top = \frac{1}{\sqrt{ND}} \begin{pmatrix} I_D \otimes V \\ E^\top \end{pmatrix}. \tag{13}$$

By an SVD decomposition, it is easy to see that, the nonzero eigenvalues of $\mathcal{G} = K^\top K$ are the same as the nonzero eigenvalues of $\Theta = KK^\top$:

$$\Theta(E, V) = \frac{1}{ND} \begin{pmatrix} I_D \otimes V^\top & E \end{pmatrix} \begin{pmatrix} I_D \otimes V \\ E^\top \end{pmatrix} = \frac{1}{ND} EE^\top + \frac{1}{ND} V^\top V I_D = e + v \cdot I_D \in \mathbb{R}^{D \times D}. \tag{14}$$

where $e, v$ are the quantities found in the main paper, and we therefore have

$$\lambda_{\max}[\mathcal{G}(E, V)] = \lambda_{\max}[\Theta(E, V)] = \lambda_{\max}\left[ \frac{1}{ND} EE^\top \right] + \frac{1}{ND} V^\top V. \tag{15}$$

---

[5]Simple application of the SVD decomposition.

**Study of $\mathcal{R}$ and of the residual.** Note that $\mathcal{R}(E, V)$ has both positive and negative eigenvalues, with spectrum symmetric along the real line. It is easy to show that

$$\lambda_{\max}[\mathcal{R}(E, V)] = -\lambda_{\min}[\mathcal{R}(E, V)] = \sqrt{\frac{\gamma^2}{ND}}\|w - w^*\|.$$

Using the fact that $\mathcal{G}$ is Hermitian, we can apply Weyl's inequality to obtain a bound on the deviation of the sharpness from the maximum eigenvalue of $\mathcal{G}$ in terms of the residual:

$$\lambda_{\max}[\Theta(E, V)] - \sqrt{\frac{\gamma^2}{ND}}\|w - w_*\|_2 \leq \lambda_{\max}[\gamma^2 \nabla^2 \mathcal{L}(E, V)] \leq \lambda_{\max}[\Theta(E, V)] + \sqrt{\frac{\gamma^2}{ND}}\|w - w_*\|_2. \tag{16}$$

Which finally yields:

$$|\lambda_{\max}[\gamma^2 \nabla^2 \mathcal{L}(E, V)] - \lambda_{\max}[\Theta(E, V)]| \leq \sqrt{\frac{\gamma^2}{ND}}\|w - w_*\|_2. \tag{17}$$

□

### C.2 Proof of Thm.5.1

We divide the proof into two parts. In the first part, we study the evolution of the unnormalized quantities $EE^\top, V^\top V$ and $EV$. Then, we study how normalization affects the dynamics.

**Part one: dynamics in a smaller space.** We go step by step recalling the gradient descent equations at the beginning of this section.

*Dynamics of $EV$.* We have :

$$\begin{aligned}
E_+ V_+ &= (E - \eta \partial_E)(V - \eta \partial_V) \\
&= EV - \eta \partial_E V - \eta E \partial_V + \eta^2 \partial_E \partial_V \\
&= EV - \frac{\eta_0}{ND} EVV^\top V + \frac{\gamma \eta_0}{\sqrt{ND}} w_* V^\top V - \frac{\eta_0}{ND} EE^\top EV + \frac{\gamma \eta_0}{\sqrt{ND}} EE^\top w_* \\
&\quad + \left( \frac{\eta_0}{ND} EVV^\top - \frac{\gamma \eta_0}{\sqrt{ND}} w_* V^\top \right) \left( \frac{\eta_0}{ND} E^\top EV - \frac{\gamma \eta_0}{\sqrt{ND}} E^\top w_* \right) \\
&= EV - \frac{\eta_0}{ND} EVV^\top V + \frac{\gamma \eta_0}{\sqrt{ND}} w_* V^\top V - \frac{\eta_0}{ND} EE^\top EV + \frac{\gamma \eta_0}{\sqrt{ND}} EE^\top w_* \\
&\quad + \frac{\eta_0^2}{N^2 D^2} EVV^\top E^\top EV - \frac{\eta_0^2 \gamma}{N^{\frac{3}{2}} D^{\frac{3}{2}}} EVV^\top E^\top w_* - \frac{\eta_0^2 \gamma}{N^{\frac{3}{2}} D^{\frac{3}{2}}} w_* V^\top E^\top EV + \frac{\gamma^2 \eta_0^2}{ND} w_* V^\top E^\top w_*
\end{aligned}$$

Let us rename

$$\tilde{w} = EV, \qquad \tilde{v} = V^\top V, \qquad \tilde{e} = EE^\top$$

then the equation becomes more compact:

$$\boxed{\begin{aligned}
\tilde{w}^+ &= \tilde{w} - \frac{\eta_0}{ND} \tilde{v}\tilde{w} + \frac{\gamma \eta_0}{\sqrt{ND}} \tilde{v} w_* - \frac{\eta_0}{ND} \tilde{e}\tilde{w} + \frac{\gamma \eta_0}{\sqrt{ND}} \tilde{e} w_* \\
&\quad + \frac{\eta_0^2}{N^2 D^2} (\tilde{w}\tilde{w}^\top)\tilde{w} - \frac{\eta_0^2 \gamma}{N^{\frac{3}{2}} D^{\frac{3}{2}}} (\tilde{w}\tilde{w}^\top) w_* - \frac{\eta_0^2 \gamma}{N^{\frac{3}{2}} D^{\frac{3}{2}}} (w_* \tilde{w}^\top)\tilde{w} + \frac{\gamma^2 \eta_0^2}{ND} (w_* \tilde{w}^\top) w_*
\end{aligned}} \tag{18}$$

Note that no quantities appear in the equation for $\tilde{w}^+$ besides $\tilde{w}, \tilde{v}, \tilde{e}$. We will see that these do not appear also in the equations for $\tilde{v}, \tilde{e}$.

*Dynamics of $V^\top V$.* Let us write down here the equations for $-\eta \partial_V$ and $-\eta \partial_V^\top$ for ease of reference:

$$-\eta \partial_V^\top = -\frac{\eta_0}{ND} V^\top E^\top E + \frac{\gamma \eta_0}{\sqrt{ND}} w_*^\top E$$

$$-\eta \partial_V = -\frac{\eta_0}{ND} E^\top EV + \frac{\gamma \eta_0}{\sqrt{ND}} E^\top w_*$$

we have

$$
\begin{aligned}
V_+^\top V_+ &= (V - \eta \partial_V)^\top (V - \eta \partial_V) \\
&= V^\top V - \eta \partial_V^\top V - \eta V^\top \partial_V + \eta^2 \partial_V^\top \partial_V \\
&= V^\top V - \frac{\eta_0}{ND} V^\top E^\top E V + \frac{\gamma \eta_0}{\sqrt{ND}} w_*^\top E V - \frac{\eta_0}{ND} V^\top E^\top E V + \frac{\gamma \eta_0}{\sqrt{ND}} V^\top E^\top w_* \\
&\quad + \left( -\frac{\eta_0}{ND} V^\top E^\top E + \frac{\gamma \eta_0}{\sqrt{ND}} w_*^\top E \right) \left( -\frac{\eta_0}{ND} E^\top E V + \frac{\gamma \eta_0}{\sqrt{ND}} E^\top w_* \right) \\
&= V^\top V - \frac{\eta_0}{ND} V^\top E^\top E V + \frac{\gamma \eta_0}{\sqrt{ND}} w_*^\top E V - \frac{\eta_0}{ND} V^\top E^\top E V + \frac{\gamma \eta_0}{\sqrt{ND}} V^\top E^\top w_* \\
&\quad + \frac{\eta_0^2}{N^2 D^2} V^\top E^\top E E^\top E V - \frac{\eta_0^2 \gamma}{N^{\frac{3}{2}} D^{\frac{3}{2}}} w_*^\top E E^\top E V - \frac{\eta_0^2 \gamma}{N^{\frac{3}{2}} D^{\frac{3}{2}}} V^\top E^\top E E^\top w_* + \frac{\gamma^2 \eta_0^2}{ND} w_*^\top E E^\top w_*.
\end{aligned}
$$

Using our notation, equations get yet again simpler:

$$
\tilde{v}^+ = \tilde{v} - 2\frac{\eta_0}{ND} \tilde{w}^\top \tilde{w} + 2\frac{\gamma \eta_0}{\sqrt{ND}} w_*^\top \tilde{w} + \frac{\eta_0^2}{N^2 D^2} \tilde{w}^\top \tilde{e} \tilde{w} - 2\frac{\eta_0^2 \gamma}{N^{\frac{3}{2}} D^{\frac{3}{2}}} w_*^\top \tilde{e} \tilde{w} + \frac{\gamma^2 \eta_0^2}{ND} w_*^\top \tilde{e} w_*.
\tag{19}
$$

Note that again no quantities appear in the equation for $\tilde{v}^+$ besides $\tilde{w}, \tilde{v}, \tilde{e}$.

*Dynamics of $E^\top E$.* For convenience, recall:

$$
-\eta \partial_E = -\frac{\eta_0}{ND} E V V^\top + \frac{\gamma \eta_0}{\sqrt{ND}} w_* V^\top
$$
$$
-\eta \partial_E^\top = -\frac{\eta_0}{ND} V V^\top E^\top + \frac{\gamma \eta_0}{\sqrt{ND}} V w_*^\top.
$$

we have

$$
\begin{aligned}
E_+ E_+^\top &= (E - \eta \partial_E)(E - \eta \partial_E)^\top \\
&= E E^\top - \eta \partial_E E^\top - \eta E \partial_E^\top + \eta^2 \partial_E \partial_E^\top \\
&= E E^\top - \frac{\eta_0}{ND} E V V^\top E^\top + \frac{\gamma \eta_0}{\sqrt{ND}} w_* V^\top E^\top - \frac{\eta_0}{ND} E V V^\top E^\top + \frac{\gamma \eta_0}{\sqrt{ND}} E V w_*^\top \\
&\quad + \left( -\frac{\eta_0}{ND} E V V^\top + \frac{\gamma \eta_0}{\sqrt{ND}} w_* V^\top \right) \left( -\frac{\eta_0}{ND} V V^\top E^\top + \frac{\gamma \eta_0}{\sqrt{ND}} V w_*^\top \right) \\
&= E E^\top - \frac{\eta_0}{ND} E V V^\top E^\top + \frac{\gamma \eta_0}{\sqrt{ND}} w_* V^\top E^\top - \frac{\eta_0}{ND} E V V^\top E^\top + \frac{\gamma \eta_0}{\sqrt{ND}} E V w_*^\top \\
&\quad + \frac{\eta_0^2}{N^2 D^2} E V V^\top V V^\top E^\top - \frac{\gamma \eta_0^2}{N^{\frac{3}{2}} D^{\frac{3}{2}}} w_* V^\top V V^\top E^\top - \frac{\gamma \eta_0^2}{N^{\frac{3}{2}} D^{\frac{3}{2}}} E V V^\top V w_*^\top + \frac{\gamma^2 \eta_0^2}{N^2 D^2} w_* V^\top V w_*^\top.
\end{aligned}
$$

Using our notation, equations get yet again simpler:

$$
\begin{aligned}
\tilde{e}^+ = \tilde{e} &- 2\frac{\eta_0}{ND} \tilde{w}\tilde{w}^\top + \frac{\gamma \eta_0}{\sqrt{ND}} w_* \tilde{w}^\top + \frac{\gamma \eta_0}{\sqrt{ND}} \tilde{w} w_*^\top \\
&+ \frac{\eta_0^2}{N^2 D^2} \tilde{v}\tilde{w}\tilde{w}^\top - \frac{\gamma \eta_0^2}{N^{\frac{3}{2}} D^{\frac{3}{2}}} \tilde{v} w_* \tilde{w}^\top - \frac{\gamma \eta_0^2}{N^{\frac{3}{2}} D^{\frac{3}{2}}} \tilde{v}\tilde{w} w_*^\top + \frac{\eta_0^2 \gamma^2}{ND} \tilde{v} w_* w_*^\top.
\end{aligned}
\tag{20}
$$

**Part two: Normalization.** Consider scalar reparameterizations.

$$
w = \alpha_w \tilde{w}, \qquad e = \alpha_e \tilde{e}, \qquad v = \alpha_v \tilde{v}
$$

While we gave the form of these normalizers already in the paper, we keep it more general here to real numbers and show that the right normalizers arise directly.

*Reparameterization of $EV$.* We have

$$w^+ = \alpha_w \tilde{w}^+$$

$$= (\alpha_w \tilde{w}) - \frac{\eta_0}{ND}\tilde{v}(\alpha_w \tilde{w}) + \frac{\gamma \eta_0 \alpha_w}{\sqrt{ND}}\tilde{v}w_* - \frac{\eta_0}{ND}\tilde{e}(\alpha_w \tilde{w}) + \frac{\gamma \eta_0 \alpha_w}{\sqrt{ND}}\tilde{e}w_*$$

$$+ \frac{\eta_0^2}{N^2 D^2}(\tilde{w}\tilde{w}^\top)(\alpha_w \tilde{w}) - \frac{\eta_0^2 \gamma}{N^{\frac{3}{2}}D^{\frac{3}{2}}}(\alpha_w \tilde{w}\tilde{w}^\top)w_* - \frac{\eta_0^2 \gamma}{N^{\frac{3}{2}}D^{\frac{3}{2}}}(w_* \tilde{w}^\top)(\alpha_w \tilde{w}) + \frac{\gamma^2 \eta_0^2}{ND}(w_*(\alpha_w \tilde{w})^\top)w_*$$

$$= (\alpha_w \tilde{w}) - \frac{\eta_0}{ND\alpha_v}(\alpha_v \tilde{v})(\alpha_w \tilde{w}) + \frac{\gamma \eta_0 \alpha_w}{\sqrt{ND}\alpha_v}(\alpha_v \tilde{v})w_* - \frac{\eta_0}{ND\alpha_e}(\alpha_e \tilde{e})(\alpha_w \tilde{w}) + \frac{\gamma \eta_0 \alpha_w}{\sqrt{ND}\alpha_e}(\alpha_e \tilde{e})w_*$$

$$+ \frac{\eta_0^2}{N^2 D^2 \alpha_w^2}(\alpha_w^2 \tilde{w}\tilde{w}^\top)(\alpha_w \tilde{w}) - \frac{\eta_0^2 \gamma}{N^{\frac{3}{2}}D^{\frac{3}{2}}\alpha_w}(\alpha_w^2 \tilde{w}\tilde{w}^\top)w_* - \frac{\eta_0^2 \gamma}{N^{\frac{3}{2}}D^{\frac{3}{2}}\alpha_w}(\alpha_w w_* \tilde{w}^\top)(\alpha_w \tilde{w}) + \frac{\gamma^2 \eta_0^2}{ND}(\alpha_w w_* \tilde{w}^\top)w_*$$

$$= w - \frac{\eta_0}{ND\alpha_v}vw + \frac{\gamma \eta_0 \alpha_w}{\sqrt{ND}\alpha_v}vw_* - \frac{\eta_0}{ND\alpha_e}ew + \frac{\gamma \eta_0 \alpha_w}{\sqrt{ND}\alpha_e}ew_*$$

$$+ \frac{\eta_0^2}{N^2 D^2 \alpha_w^2}ww^\top w - \frac{\eta_0^2 \gamma}{N^{\frac{3}{2}}D^{\frac{3}{2}}\alpha_w}(ww^\top)w_* - \frac{\eta_0^2 \gamma}{N^{\frac{3}{2}}D^{\frac{3}{2}}\alpha_w}(w_* w^\top)w + \frac{\gamma^2 \eta_0^2}{ND}(w_* w^\top)w_*.$$

We would like $\alpha_w, \alpha_e, \alpha_v$ to be such that on the right-hand side we have no width dependency. To do that we need (first and second line refer to first and second lines in the equation)

$$\alpha_v \propto \frac{1}{N}, \qquad \alpha_w \propto \alpha_v \frac{\sqrt{N}}{\gamma}, \qquad \alpha_e \propto \frac{1}{N}, \qquad \alpha_w \propto \alpha_e \frac{\sqrt{N}}{\gamma}$$

$$\alpha_w \propto \frac{1}{N}, \qquad \alpha_w \propto \frac{\gamma}{N^{\frac{3}{2}}D^{\frac{3}{2}}}, \qquad \gamma^2 \propto N$$

where proportionality can depend on any factor (e.g. $d$) except of course $N$. Crucially note that the equations require $\gamma \propto \sqrt{N}$. **So this can be done for $\mu P$ but not for NTP (except if $d \simeq N$).** Further, we need

$$\alpha_w, \alpha_e, \alpha_v \propto \frac{1}{N}.$$

To get that $w \to w_*$, we choose (as in the main paper)

$$\alpha_w = \frac{1}{\gamma \sqrt{ND}} = \frac{1}{N\sqrt{D}} \text{ (for } \mu P\text{)}, \qquad \alpha_e = \frac{1}{ND}, \qquad \alpha_v = \frac{1}{ND}.$$

So, we get

$$w^+ = w - \frac{\eta_0}{ND\alpha_v}vw + \frac{\gamma \eta_0 \alpha_w}{\sqrt{ND}\alpha_v}vw_* - \frac{\eta_0}{ND\alpha_e}ew + \frac{\gamma \eta_0 \alpha_w}{\sqrt{ND}\alpha_e}ew_*$$

$$+ \frac{\eta_0^2}{N^2 D^2 \alpha_w^2}ww^\top w - \frac{\eta_0^2 \gamma}{N^{\frac{3}{2}}D^{\frac{3}{2}}\alpha_w}(ww^\top)w_* - \frac{\eta_0^2 \gamma}{N^{\frac{3}{2}}D^{\frac{3}{2}}\alpha_w}(w_* w^\top)w + \frac{\gamma^2 \eta_0^2}{ND}(w_* w^\top)w_*$$

$$= w - \eta_0 vw + \eta_0 vw_* - \eta_0 ew + \eta_0 ew_*$$

$$+ \frac{\eta_0^2 \gamma^2}{ND}ww^\top w - \frac{\eta_0^2 \gamma^2}{ND}(ww^\top)w_* - \frac{\eta_0^2 \gamma^2}{ND}(w_* w^\top)w + \frac{\gamma^2 \eta_0^2}{ND}(w_* w^\top)w_*$$

*Reparameterization of $V^\top V$.* Recall that

$$\tilde{v}^+ = \tilde{v} - 2\frac{\eta_0}{ND}\tilde{w}^\top \tilde{w} + 2\frac{\gamma \eta_0}{\sqrt{ND}}w_*^\top \tilde{w} + \frac{\eta_0^2}{N^2 D^2}\tilde{w}^\top \tilde{e}\tilde{w} - 2\frac{\eta_0^2 \gamma}{N^{\frac{3}{2}}D^{\frac{3}{2}}}w_*^\top \tilde{e}\tilde{w} + \frac{\gamma^2 \eta_0^2}{ND}w_*^\top \tilde{e}w_*$$

This implies, after scaling

$$v^+ = \alpha_v \tilde{v}^+$$

$$= (\alpha_v \tilde{v}) - 2\frac{\eta_0 \alpha_v}{ND} \tilde{w}^\top \tilde{w} + 2\frac{\gamma \eta_0 \alpha_v}{\sqrt{ND}} w_*^\top \tilde{w}$$

$$+ \frac{\eta_0^2 \alpha_v}{N^2 D^2} \tilde{w}^\top \tilde{e}\tilde{w} - 2\frac{\eta_0^2 \gamma \alpha_v}{N^{\frac{3}{2}} D^{\frac{3}{2}}} w_*^\top \tilde{e}\tilde{w} + \frac{\gamma^2 \eta_0^2 \alpha_v}{ND} w_*^\top \tilde{e}w_*$$

$$= v - 2\frac{\eta_0 \alpha_v}{ND\alpha_w^2} w^\top w + 2\frac{\gamma \eta_0 \alpha_v}{\sqrt{ND}\alpha_w} w_*^\top w$$

$$+ \frac{\eta_0^2 \alpha_v}{N^2 D^2 \alpha_w^2 \alpha_e} w^\top ew - 2\frac{\eta_0^2 \gamma \alpha_v}{N^{\frac{3}{2}} D^{\frac{3}{2}} \alpha_w \alpha_e} w_*^\top ew + \frac{\gamma^2 \eta_0^2 \alpha_v}{ND\alpha_e} w_*^\top ew_*$$

It is easy to see that the choice $\alpha_w, \alpha_e, \alpha_v \propto \frac{1}{N}$ in addition with $\gamma = \sqrt{N}$ gives independence of the RHS to width.

Under our choices

$$\alpha_w = \frac{1}{\gamma\sqrt{ND}} = \frac{1}{N\sqrt{D}} \text{ (for } \mu P), \qquad \alpha_e = \frac{1}{ND}, \qquad \alpha_v = \frac{1}{ND},$$

we get

$$v^+ = v + \frac{\eta_0 \gamma^2}{ND} \left[ -2w^\top w + 2w_*^\top w \right] + \frac{\eta_0^2 \gamma^2}{ND} \left[ w^\top ew - 2w_*^\top ew + w_*^\top ew_* \right]$$

*Reparameterization of $EE^\top$.* Let's substitute the scaled version in the equations

$$e^+ = \alpha_e \tilde{e}^+$$

$$= e - 2\frac{\eta_0 \alpha_e}{ND\alpha_w^2} ww^\top + \frac{\gamma \eta_0 \alpha_e}{\sqrt{ND}\alpha_w} w_* w^\top + \frac{\gamma \eta_0 \alpha_e}{\sqrt{ND}\alpha_w} ww_*^\top$$

$$+ \frac{\eta_0^2 \alpha_e}{N^2 D^2 \alpha_w^2 \alpha_v} vww^\top - \frac{\gamma \eta_0^2 \alpha_e}{N^{\frac{3}{2}} D^{\frac{3}{2}} \alpha_v \alpha_w} vw_* w^\top - \frac{\gamma \eta_0^2 \alpha_e}{N^{\frac{3}{2}} D^{\frac{3}{2}} \alpha_v \alpha_w} vww_*^\top + \frac{\eta_0^2 \gamma^2 \alpha_e}{ND\alpha_v} vw_* w_*^\top.$$

With our choices

$$\alpha_w = \frac{1}{\gamma\sqrt{ND}} = \frac{1}{N\sqrt{D}} \text{ (for } \mu P), \qquad \alpha_e = \frac{1}{ND}, \qquad \alpha_v = \frac{1}{ND}.$$

we get

$$e^+ = e + \frac{\eta_0 \gamma^2}{ND} \left[ -2ww^\top + w_* w^\top + ww_*^\top \right] + \frac{\eta_0^2 \gamma^2}{ND} \left[ vww^\top - vw_* w^\top - vww_*^\top + vw_* w_*^\top \right]$$

### C.2.1 Initialization

**Proposition C.1.** *At initialization, as $N \to \infty$, $e \xrightarrow{\mathbb{P}} e^\infty := \frac{1}{D} I_D$ and $v \xrightarrow{\mathbb{P}} v^\infty := \frac{1}{D}$. Moreover, errors from $\infty-$ initialization scale as $\frac{1}{\sqrt{N}}$ in expectation: $\mathbb{E}|v - v^\infty|^2, \mathbb{E}|e_{ij} - e_{i,j}^\infty|^2 \leq \frac{2}{ND}, \forall i, j \in [D]$. While for $\gamma = 1$ (NTP) $w$ at initialization is in the limit Gaussian with elementwise variance $1/D$, for $\gamma = \sqrt{N}$ ($\mu P$) we have $w \xrightarrow{\mathbb{P}} w^\infty := 0$, with elementwise variations scaling as $\frac{1}{\sqrt{N}}$: $\mathbb{E}|w_i - w_i^\infty|^2 = \frac{1}{ND}, \forall i \in [D]$.*

First, note that trivially

$$\mathbb{E}[w_i] = 0,$$

and

$$\mathbb{E}[w_i]^2 = \frac{1}{\gamma^2 ND} \sum_{j,j'=1}^{N} \mathbb{E}[E_{ij} E_{ij'} V_j V_{j'}] = \frac{1}{D\gamma^2}.$$

Next:

$$\mathbb{E}[e_{ij}] = \frac{1}{ND} \sum_{k=1}^{N} \mathbb{E}[E_{ik}E_{jk}] = \frac{1}{D}\delta_{ij},$$

and

$$\mathbb{E}[e_{ij}^2] = \frac{1}{N^2D^2} \sum_{k,k'=1}^{N} \mathbb{E}[E_{ik}E_{jk}E_{ik'}E_{jk'}].$$

For $i \neq j$

$$\mathbb{E}[e_{ij}^2] = \frac{1}{N^2D^2} \sum_{k,k'=1}^{N} \mathbb{E}[E_{ik}E_{ik'}]\mathbb{E}[E_{jk}E_{jk'}] = \frac{1}{ND^2} \qquad (i \neq j)$$

For $i = j$

$$\mathbb{E}[e_{ij}^2] = \frac{1}{N^2D^2} \sum_{k,k'=1}^{N} \mathbb{E}[E_{ik}^2 E_{ik'}^2]$$

$$= \frac{1}{N^2D^2} \sum_{k \neq k'} \mathbb{E}[E_{ik}^2]\mathbb{E}[E_{ik'}^2] + \frac{1}{N^2D^2} \sum_{k=1}^{N} \mathbb{E}[E_{ik}^4]$$

$$= \frac{N(N-1)}{N^2D^2} + \frac{3}{ND^2}$$

$$= \frac{N+2}{ND^2} \qquad (i = j).$$

So

$$\mathrm{Var}[e_{ij}] = \frac{N+2}{ND^2} - \frac{1}{D^2} = \frac{2}{ND^2}.$$

Finally:

$$\mathbb{E}[v] = \frac{1}{ND} \sum_{i=1}^{N} \mathbb{E}[V_iV_i] = \frac{1}{D},$$

and

$$\mathbb{E}[v^2] = \frac{1}{N^2D^2} \sum_{i,i'=1}^{N} \mathbb{E}[V_iV_iV_{i'}V_{i'}] = \frac{1}{N^2D^2} \sum_{i \neq i'}^{N} \mathbb{E}[V_i^2]\mathbb{E}[V_{i'}^2] + \frac{1}{N^2D^2} \sum_{i=1}^{N} \mathbb{E}[V_i^4] = \frac{N-1}{ND^2} + \frac{3}{ND^2} = \frac{N+2}{ND^2}.$$

So

$$\mathrm{Var}[v] = \frac{2}{ND^2}.$$

### C.2.2   Proof of Prop. 5.3

**EoS Basic Derivation on quadratic (global linear dynamics).**   In quadratic potentials $\frac{1}{2}w^\top H w$, we are at EOS if and only if $\eta = 2/\lambda_{\max}(H)$. The proof of this can be translated into a Jacobian argument on the gradient update map. The update map here is $w^+ = (I - \eta H)w$, which has Jacobian $I - \eta H$. The eigenvalues of this map are in the range $[1 - \lambda_{\max}(H)\eta, 1 - \lambda_{\min}(H)\eta]$, where bounds are tight. For asymptotic stability (i.e. we converge), we have the necessary condition $\lambda_{\max}(H) < 2/\eta$, which guarantees all eigenvalues are in the range $(-1, 1)$. To be at edge of stability means $\lambda_{\max}(H) = 2/\eta$, that is if and only if the Jacobian has eigenvalues in $[-1, 1]$, with an eigenvalue equal to $-1$.

**Remark: What changes here?**   We are dealing with a nonlinear transition map where the Jacobian has explicit $O(\eta^2)$ terms. To be precise in this setting, we need to carefully study the Jacobian and cannot assume its first-order approximation in $\eta_0$, i.e. $I - \eta_0 \nabla^2 L$, is accurate.

**Proof.** We need to compute the Jacobian for the dynamical system $G : (w, e, v) \to (w^+, e^+, v^+)$. Note that – specifically at $w = w^*$,

$$\frac{\partial w^+}{\partial w}\Big|_{w=w^*} = I_D - \eta_0(e + vI) + \frac{\eta_0^2\gamma^2}{ND}w_* \otimes w_*^\top, \qquad \frac{\partial w^+}{\partial e}\Big|_{w=w^*} = 0, \quad \frac{\partial w^+}{\partial v}\Big|_{w=w^*} = 0 \tag{21}$$

$$\frac{\partial e^+}{\partial w}\Big|_{w=w^*} = \star, \qquad\qquad\qquad \frac{\partial e^+}{\partial e}\Big|_{w=w^*} = I_{D^2}, \quad \frac{\partial e^+}{\partial v}\Big|_{w=w^*} = 0 \tag{22}$$

$$\frac{\partial v^+}{\partial w}\Big|_{w=w^*} = \star, \qquad\qquad\qquad \frac{\partial v^+}{\partial e}\Big|_{w=w^*} = \star, \quad \frac{\partial v^+}{\partial v}\Big|_{w=w^*} = 1 \tag{23}$$

where we do not care about the values with a $\star$ because anyway the resulting matrix is lower-triangular:

$$J_G(w_*, e_*, v_*) = \begin{pmatrix} I - \eta_0(v_*I + e_*) + \frac{\eta_0^2\gamma^2}{ND}w_* \otimes w_*^\top & 0 & 0 \\ \star & I & 0 \\ \star & \star & I \end{pmatrix}$$

As in the quadratic setting, to be at the edge of stability, we require this matrix to have eigenvalues in $[-1, 1]$, with at least an eigenvalue precisely equal to $-1$. Without loss in generality, we discuss eigenvalues in descending order for each one of our matrices:

$$\lambda_1 \geq \lambda_2 \geq \cdots \geq \lambda_D.$$

Note that $J_G(w_*, e_*, v_*)$ is block lower-triangular, hence its eigenvalues are

$$\lambda(J_G(w_*, e_*, v_*)) = \lambda\left(I - \eta_0(v_*I + e_*) + \frac{\eta_0^2\gamma^2}{ND}w_* \otimes w_*^\top\right) \cup \{1\}.$$

The necessary condition for being at the edge of stability is then

$$\lambda_D\left(I - \eta_0\Theta^* + \frac{\eta_0^2\gamma^2}{ND}w_* \otimes w_*^\top\right) = -1.$$

where $\Theta^* := v_*I + e_*$. We then ask the question: for which combination of $\Theta^*$ and $\eta_0$s can this happen? Note that for any eigenvalue index $k$ (eigenvalues sorted in descending order), $\eta_0 \geq 0$ and $\Theta^* \succeq 0$:

$$\lambda_k(I - \eta_0\Theta^*) = 1 - \eta_0\lambda_{D-k}(\Theta^*),$$

Thus:

$$\lambda_D(I - \eta_0\Theta^*) = 1 - \eta_0\lambda_1(\Theta^*), \qquad \lambda_1(I - \eta_0\Theta^*) = 1 - \eta_0\lambda_D(\Theta^*).$$

Let us set $\Gamma^* = I - \eta_0\Theta^*$ – a matrix of which we know the eigenvalues. The matrix we want to study is a rank-1 $O(\eta_0^2)$ perturbation: $\Gamma^* + P$ where $P = \frac{\eta_0^2\gamma^2}{ND}w_* \otimes w_*^\top$. Intuitively, this perturbation matrix $P$ does not modify much the eigenvalues.

To start, note that the CourantFischer Theorem [91] implies (see e.g. [92], exercise 10.2.2.):

$$\lambda_D(\Gamma^*) \leq \lambda_D(\Gamma^* + P)$$

Next, recall Weyl's inequality for Hermitian (in this case, symmetric) matrices:

$$\lambda_{i+j-1}(\Gamma^* + P) \leq \lambda_i(\Gamma^*) + \lambda_j(P)$$

Apply this to $j = 1, i = D$, we get

$$\lambda_D(\Gamma^* + P) \leq \lambda_D(\Gamma^*) + \lambda_1(P).$$

All in all, we proved the bound

$$\lambda_D(\Gamma^*) \leq \lambda_D(\Gamma^* + P) \leq \lambda_D(\Gamma^*) + \lambda_1(P).$$

Notice that this implies, given the structure of $P$,

$$\lambda_D(\Gamma^*) \le \lambda_D(\Gamma^* + P) \le \lambda_D(\Gamma^*) + \frac{\eta_0^2 \gamma^2}{ND}\|w_*\|^2.$$

For EoS, we require $\lambda_D(\Gamma_* + P) = -1$. Hence we necessarily need

$$\lambda_D(\Gamma^*) \in \left[-1 - \frac{\eta_0^2 \gamma^2}{ND}\|w_*\|^2, -1\right].$$

Recall that $\lambda_D(\Gamma^*) = \lambda_D(I - \eta_0 \Theta^*) = 1 - \eta_0 \lambda_1(\Theta^*)$, hence the condition is

$$-1 - \frac{\eta_0^2 \gamma^2}{ND}\|w_*\|^2 \le 1 - \eta_0 \lambda_1(\Theta^*) \le -1.$$

This implies

$$-2 - \frac{\eta_0^2 \gamma^2}{ND}\|w_*\|^2 \le -\eta_0 \lambda_1(\Theta^*) \le -2.$$

Dividing everything by $\eta_0$,

$$-\frac{2}{\eta_0} - \frac{\eta_0 \gamma^2}{ND}\|w_*\|^2 \le -\lambda_1(\Theta^*) \le -\frac{2}{\eta_0},$$

hence

$$\frac{2}{\eta_0} \le \lambda_1(\Theta^*) \le \frac{2}{\eta_0} + \frac{\eta_0 \gamma^2}{ND}\|w_*\|^2$$

To conclude, note that $\lambda_1(\Theta^*)$ coincides with the Hessian $\lambda_1$.

*Note on marginal stability (sufficient conditions).* At the same time, for stability, we need (sufficient condition)

$$-1 \le \lambda_D(\Gamma^* + P) \le \lambda_1(\Gamma^* + P) \le 1.$$

The condition $-1 \le \lambda_D(\Gamma^* + P)$ is implied by $-1 \le \lambda_D(\Gamma^*)$ (Courant-Fischer). This yields

$$-1 \le 1 - \eta_0 \lambda_1(\Theta^*),$$

that is

$$\eta_0 \le \frac{2}{\lambda_1(\Theta^*)}.$$

Next, we need $\lambda_1(\Gamma^* + P) \le 1$. A sufficient condition for this is (again by Weyl)

$$\lambda_1(\Gamma^*) + \lambda_1(P) \le 1,$$

which implies

$$\lambda_1(\Gamma^*) \le 1 - \frac{\eta_0^2 \gamma^2}{ND}\|w_*\|^2.$$

That is

$$1 - \eta_0 \lambda_D(\Theta^*) \le 1 - \frac{\eta_0^2 \gamma^2}{ND}\|w_*\|^2,$$

i.e.

$$\frac{\eta_0^2 \gamma^2}{ND}\|w_*\|^2 \le \eta_0 \lambda_D(\Theta^*),$$

that is

$$\eta_0 \le \frac{\lambda_D(\Theta^*)ND}{\gamma^2 \|w_*\|^2}.$$

All in all a **sufficient** condition for marginal stability is

$$\eta_0 \le \min\left\{\frac{2}{\lambda_1(\Theta^*)}, \frac{\lambda_D(\Theta^*)ND}{\gamma^2 \|w_*\|^2}\right\}.$$

The bound is not vacuous in the case of $N \ge D$ and different data points.

**Note.** What we provided above is a sufficient condition for marginal stability (all eigenvalues in $[-1, 1]$). This is derived just for interest of the reader but does not affect the result: indeed, a sufficient condition can be not tight (as likely the case here). What we instead proved is that if we are at EoS, than necessarily (i.e., in every instance of the problem) $\frac{2}{\eta_0} \leq \lambda_1(\Theta^*) \leq \frac{2}{\eta_0} + \frac{\eta_0^2 \gamma^2}{ND} \|w_*\|^2$.

### C.3  Theory for *Standard Parametrization* (SP)

As in $\mu$P and NTP, we consider the simplified loss

$$L(E, V) = \frac{1}{2} \|EV - w^*\|^2 \tag{24}$$

where $E \in \mathbb{R}^{D \times N}, V \in \mathbb{R}^{N \times 1}$. Compared to $\mu$P and NTP, normalizing factors appear in the initialization: $E_{ij} \sim \mathcal{N}(0, 1/D)$, $V_j \sim \mathcal{N}(0, 1/N)$, $\forall i, j$. Gradients are $\partial_E = EVV^\top - w_* V^\top$, $\partial_V = E^\top EV - E^\top w_*$.

#### C.3.1  Dynamics equations, same form as $\mu P$ and NTP

Let us rename $w = EV$, $v = V^\top V$, $e = EE^\top$, then the dynamics gradient descent with stepsize $\eta$ becomes, in $(w, e, v) \in \mathbb{R}^{D+D^2+1}$ space:

$$w^+ = w - \eta(v \cdot I_D + e)(w - w_*) + \eta^2(ww^\top - w_* w^\top)(w - w_*).$$
$$e^+ = e + \eta \left[ -2ww^\top + w_* w^\top + ww_*^\top \right] + \eta^2 \left[ vww^\top - vw_* w^\top - vww_*^\top + vw_* w_*^\top \right].$$
$$v^+ = v + \eta \left[ -2w^\top w + 2w_*^\top w \right] + \eta^2 \left[ w^\top ew - 2w_*^\top ew + w_*^\top ew_* \right].$$

Hence, under SP, $(w, e, v)$ at different widths have width-independent evolution laws in $\mathbb{R}^{D+D^2+1}$ – same happens under $\mu$ P. **However**, in contrast to $\mu$P, initialization of this system does not converge as $N \to \infty$. Hence, actual dynamics across widths are drastically different, as we will see next.

#### C.3.2  Initialization

First, note that trivially $\mathbb{E}[w_i] = 0$, and $\mathbb{E}[w_i]^2 = \sum_{j, j'=1}^N \mathbb{E}[E_{ij} E_{ij'} V_j V_{j'}] = \sum_{j=1}^N \mathbb{E}[E_{ij}^2 V_j^2] = \frac{1}{D}$. This makes sense since the forward pass is normalized. However, at initialization:

$$\mathbb{E}[e_{ij}] = \sum_{k=1}^N \mathbb{E}[E_{ik} E_{jk}] = \frac{N}{D} \delta_{ij},$$

that already **includes a dependency on the width** $N$, further: $\mathbb{E}[e_{ij}^2] = \sum_{k,k'=1}^N \mathbb{E}[E_{ik} E_{jk} E_{ik'} E_{jk'}]$. Hence

$$\mathbb{E}[e_{ij}^2] = \sum_{k,k'=1}^N \mathbb{E}[E_{ik} E_{ik'}] \mathbb{E}[E_{jk} E_{jk'}] = \sum_{k=1}^N \mathbb{E}[E_{ik}^2] \mathbb{E}[E_{jk}^2] = \frac{N}{D^2} \qquad (i \neq j),$$

$$\mathbb{E}[e_{ij}^2] = \sum_{k,k'=1}^N \mathbb{E}[E_{ik}^2 E_{ik'}^2] = \sum_{k \neq k'} \mathbb{E}[E_{ik}^2] \mathbb{E}[E_{ik'}^2] + \sum_{k=1}^N \mathbb{E}[E_{ik}^4] = \frac{N(N-1)}{D^2} + \frac{3N}{D^2} = \frac{N(N+2)}{D^2} \qquad (i = j).$$

So $\mathrm{Var}[e_{ij}] = O\left(\frac{N}{D^2}\right)$. Finally:

$$\mathbb{E}[v] = \sum_{i=1}^N \mathbb{E}[V_i V_i] = 1,$$

$$\mathbb{E}[v^2] = \sum_{i,i'=1}^N \mathbb{E}[V_i V_i V_{i'} V_{i'}] = \sum_{i \neq i'} \mathbb{E}[V_i^2] \mathbb{E}[V_{i'}^2] + \sum_{i=1}^N \mathbb{E}[V_i^4] = \frac{N(N-1)}{N^2} + \frac{3N}{N^2} = \frac{N(N+2)}{N^2}.$$

So $\mathrm{Var}[v] = O\left(\frac{1}{N^2}\right)$.

### C.3.3  Conclusion

While the laws $(w, e, v) \rightarrow (w^+, e^+, v^+)$ are not width dependent in SP, $e$ has initialization of scale $O(N)$. As such, while the forward pass (i.e. $w$) is $O(1)$ at initialization, the trajectory of $(w, e, v)$ under gradient descent starts at a width-dependent point $(O(1), O(N), O(1))$; hence it is drastically different at different widths. Further, the NTK $(v \cdot I_D + e)$ is also $O(N)$, and this controls the evolution of the forward pass $w$: $w^+ = w - \eta(v \cdot I_D + e)(w - w_*) + O(\eta^2)$. We therefore validate also in this simple setting the derivation by [5]: if $\eta = O(1)$, forward pass of SP blows up after one step of gradient descent.

## D  Connection between the eigenvalues of the Hessian and NTK matrix

Recall that for MSE loss and one-dimensional output $f(x)$, the Gauss-Newton decomposition of the Hessian $H$ reads:

$$H = \mathcal{G} + \mathcal{R} = \sum_{i=1}^{|\mathcal{D}|} \nabla_\theta f(x_i) \nabla_\theta f(x_i)^\top + \sum_{i=1}^{|\mathcal{D}|} \nabla_\theta^2 f(x_i)(y_i - f(x_i)), \tag{25}$$

where $\mathcal{G}$ is the Gaussian-Newton matrix. This can be generalized to (1) different loss functions and (2) multidimensional output $f(x) \in \mathbb{R}^k$, where $k$ is the dimension of the logits (i.e. the number of classes in classification problems). Here we notice that in (2) we exactly preserve the connection between GGN and NTK's spectra, while in (1) we have an extra term in $\mathcal{G}$ that causes a deviation from the exact correspondence. However, we show that this deviation is largely negligible in practice, in the sense that $\mathcal{G}$ will have the same spectrum as the NTK as training progresses, and $\mathcal{G}$ still dominates $\mathcal{R}$, as one would expect from the experiments in the main text (e.g Fig. 1). We begin by defining the Gauss-Newton matrix (GN) in the case of cross-entropy loss, following [12]:

$$\mathcal{G} = \sum_{i=1}^{|\mathcal{D}|} \nabla_\theta f(x_i) \bar{H}_\mathcal{L} \nabla_\theta f(x_i)^\top$$

where now $\nabla_\theta f(x_i) \in \mathbb{R}^{kP}$, and $\bar{H}_\mathcal{L} \in \mathbb{R}^{kP \times kP}$ is a block-diagonal matrix where the $k \times k$ Hessian of the loss ($H_\mathcal{L}$) with respect to model output is repeated $P$ times. Again, we can stack the Jacobian vectors $\nabla_\theta f(x_i)$ for all the datapoints into $K \in \mathbb{R}^{|\mathcal{D}| \times kP}$, thus:

$$\mathcal{G} = K^\top \bar{H}_\mathcal{L} K \tag{26}$$

For MSE loss, $\bar{H}_\mathcal{L}$ is the identity, hence the correspondence to the NTK matrix is maintained (same sharpness). However, for the cross-entropy loss, the first derivative of the loss with respect to the model output $\Delta := \nabla_{f(x)} \mathcal{L}$ can be shown to be $\Delta = \sigma(f(x)) - y$, where $y$ is the one hot vector encoding the true classes, and $\sigma(\cdot)$ denotes the softmax activation. Hence, for the Hessian, we have:

$$[H_\mathcal{L}]_{ij} = \delta_{ij}\sigma(f(x))_i - \sigma(f(x))_i \sigma(f(x))_j, \tag{27}$$

which in general deviates from the identity. However, during training the model increase the probability of correct predictions, and thus $H_\mathcal{L}$ gets closer to the identity, thus having an asymptotically negligible effect on the Gauss-Newton matrix $\mathcal{G}$ and its correspondence to the NTK.

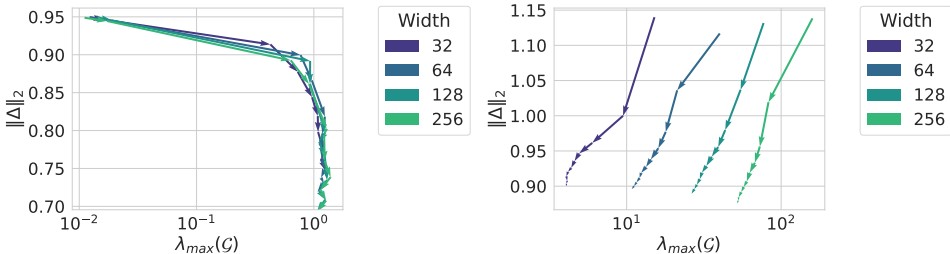

Figure 11: Norm of the residual and top eigenvalue of the GN matrix, where the vector field shows the evolution of these quantities during training for a fixed learning rate. Left: $\mu$P- note that all curves have a sharpening phase, after which the residual continues to decrease. Right: NTP - Increasing the width reduces the sharpening, since it approaches the infinite width limit where the NTK matrix becomes asymptotically constant.

We now perform experiments to test whether $\mathcal{G}$ dominates the residual for cross-entropy loss, in order to support our claim on the connection between feature learning and optimization. We plot the evolution of the largest eigenvalue of the GN matrix and the residual norm through time in Figure 11 for $\mu$P and NTP. The largest eigenvalue of this matrix is computed using a power iteration algorithm based on the implementation provided in [93]. Note that while for $\mu$P we observe a large amount of progressive sharpening during training, for NTP the sharpness becomes asymptotically constant. The architecture used for the plot is a convolutional neural network as described in J, trained on CIFAR-10 for 10 epochs using cross-entropy loss. These experiments confirm the results obtained for MSE loss in the main text (Fig. 5).

# E   Late-time dynamics and batch size ablations

## E.1   Late-time dynamics

It was noted in [16] (Appendix C) that with cross-entropy loss (adopted here), the sharpness decreases towards the end of training. Here in Fig. 12, we show that while the dynamics are remarkably consistent during the first part of training, they diverge during the phase transition in which the sharpness begins to drop, with wider models starting to exhibit the sharpness drop earlier. Hence, this transition point is highly width-dependent, and it coincides with a slight shift in the optimal learning rate. This late phase happens when the classifier maximizes the margin between classes [16], and can be largely prevented by using data augmentation techniques, as we exemplify in Sec.E.2.

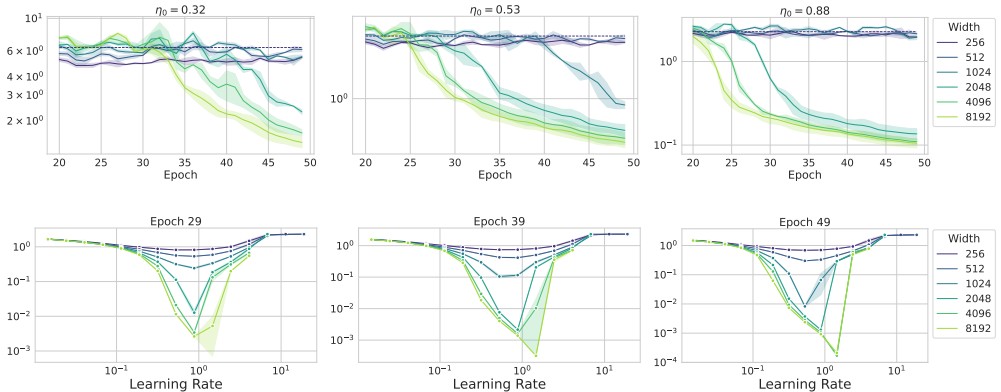

Figure 12: Late-time dynamics. We study the same setting as in Fig. 26 (for $\mu$P), under longer training time. Notice how when training longer, the sharpness decreases once the model gets closer to convergence. In this phase, there is a shift of the optimal learning rate, as the bottom row shows.

## E.2   The effect of Batch Size and Data Augmentation

**Batch Size**   We test what happens to the sharpness and optimal learning rate when the batch size is increased. The sharpness is well-known to increase with the batch size, as it is shown also in Cohen et al. [16] and Jastrzkebski et al. [31].

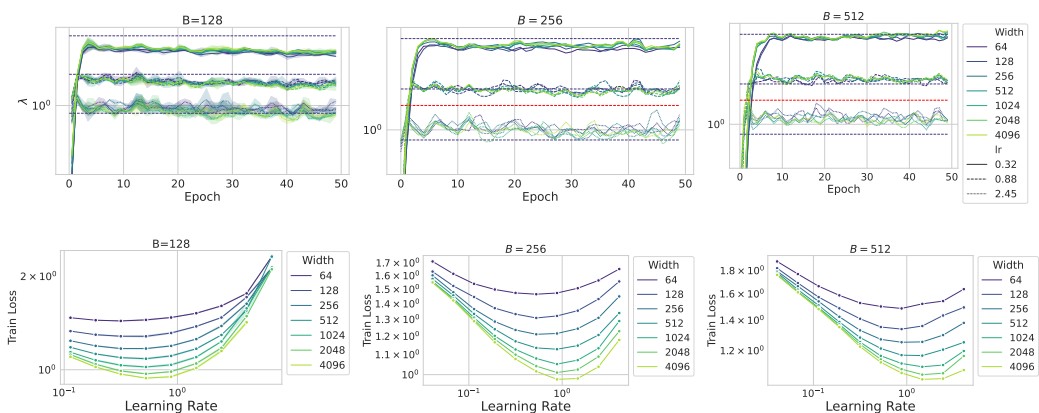

Figure 13: Batch size ablation on a three layer convolutional network. The red dashed line indicates the sharpness of $4/\eta_0$, only shown for the largest learning rate where the sharpness rises above the EoS threshold. Parameters: dataset: CIFAR-10, with data augmentation, epochs $= 50$. The learning rate shown above are: $(0.32, 0.88, 2.45)$

Here we add that under $\mu$P, the batch size we tested $(128, 256, 512)$ do not influence significantly the width-independence phenomenon of the sharpness, as we summarize in Fig. 13. We observe good

learning rate transfer across all the tested batch sizes. We also observe that the optimal learning rate increases with the batch sizes by roughly a factor of 2.

**Data Augmentation** We repeat the same experiment, varying the batch size, but this time we turn on data augmentation using standard transformations (random crops, horizontal flips, 10-degrees rotations). The results are in Fig. 14 (to compare with Fig. 13). Notice how data augmentation has a stabilizing effect on the sharpness, delaying the late phase of training where the sharpness drops, as analyzed in Sec. E.1 [16, 28]. On the other hand, Thus, under regularization techniques such as data augmentation, we should expect better hyperparameter transfer.

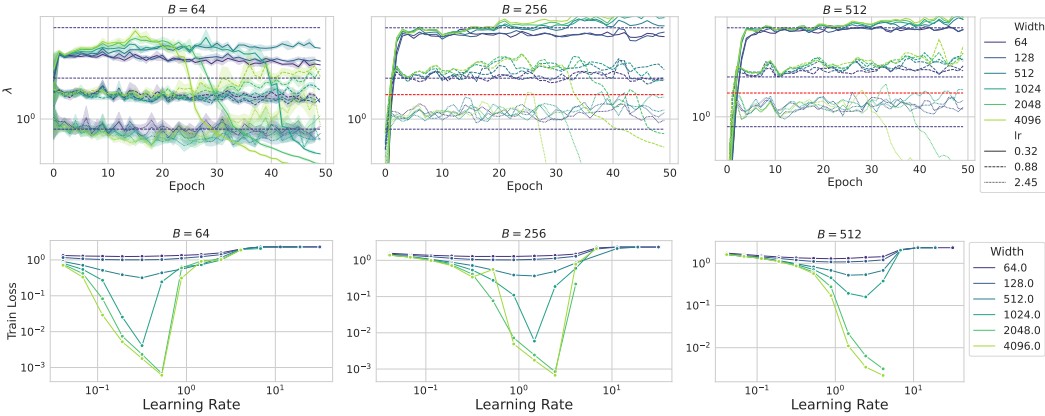

Figure 14: Batch size ablation. The red dashed line indicates the sharpness of $4/\eta_0$, only shown for the largest learning rate where the sharpness rises above the EoS threshold. Parameters: dataset: CIFAR-10, without data augmentation, epochs $= 50$. The learning rate shown above are: $(0.32, 0.88, 2.45)$

# F   Large-Scale experiments, more Datasets and Optimizers

We provide empirical validation of our findings and show that in realistic architectures, such as Transformers (ViTs, GPT-2) and ResNets, we achieve width (or depth, respectively) independent sharpness. These results empirically demonstrate that our findings extend to different modalities and architectures.

## F.1   GPT-2 experiments on WikiText

Figure 15 shows the transfer of a GPT-2 model trained on WikiText for 40 epochs with Depth-$\mu$P and Adam, without learning rate scaling. A similar plot but for $\mu$P (without the depth parameterization) is presented in Figure 16. Note that the sharpness exhibits a width independent behaviour.

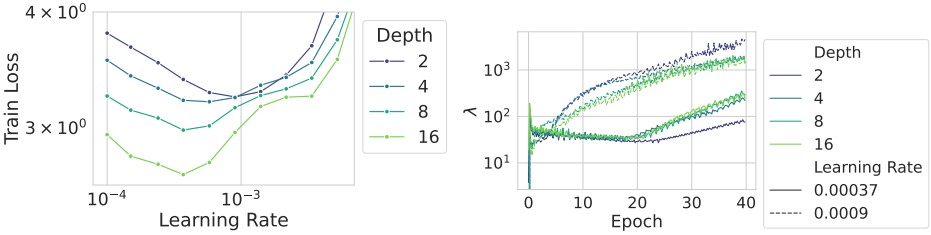

Figure 15: GPT-2 trained on WikiText using Adam, with fixed learning rate, width $512$. Note that due to the structure of the transformer block containing $k \geq 2$ layers per block, the transfer is imperfect and thus the sharpness is also not super consistent. Here, in contrast to Depth-$\mu$P's prescription in Table 1, we do not rescale the learning rate by $1/\sqrt{L}$. See also App. J.

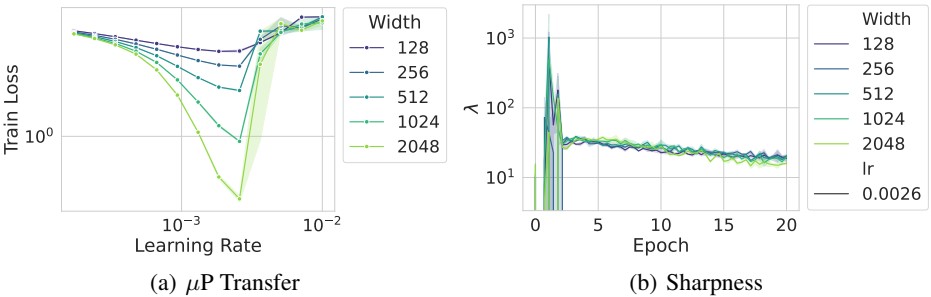

(a) $\mu$P Transfer

(b) Sharpness

Figure 16: Post-LN Transformers (similar to GPT-2) trained with Adam on WikiText-2 parameterized with $\mu$P showing learning transfer in width (a) and super consistent sharpness evolution (b). HPs: 2 layers, 2 heads, 20 epochs, batch size 512, 100 warmup steps, sequence length 35.

## F.2 ConvNets Experiments on Larger Datasets and Adam(W)

Figures 17 and 18 show residual ConvNets parameterized with $\mu$P and Depth-$\mu$P respectively trained on TinyImagenet, and Figure 19 shows the evolution of a similar model parameterized with $\mu$P trained on Imagenet. Similarly, we study the evolution of residual ConvNets when trained with Adam on CIFAR-10 with 1 and 2 layers per block in Figure 20. Finally, we show the results of training residual ConvNets under $\mu$P with Adam and AdamW in Figure 21.

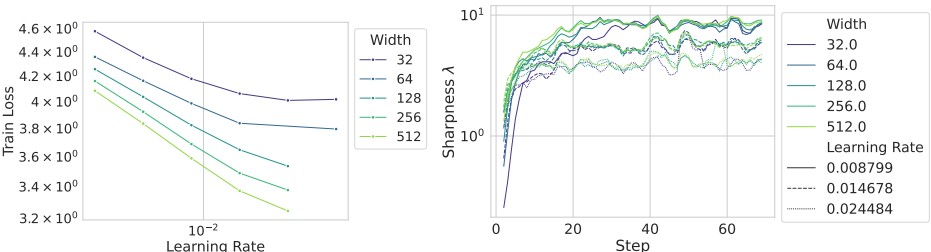

Figure 17: Residual convolutional networks (ResNets) trained on Tiny-ImageNet with stochastic gradient descent. Left figure shows the learning rate transfers across width in ResNets parameterized with $\mu$P. Right figure shows that for a fixed learning rate, the sharpness becomes width independent during training. Parameters: batch size 64, epochs 10.

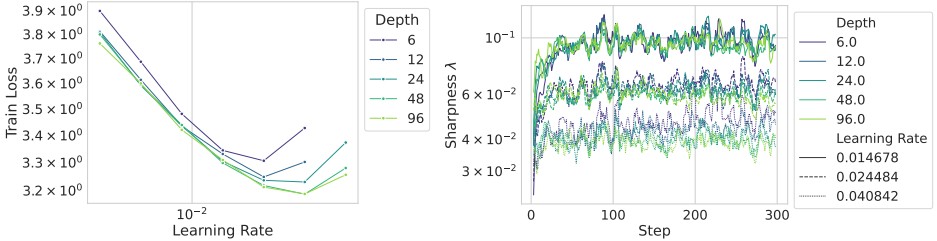

Figure 18: Residual convolutional networks trained on Tiny-ImageNet with stochastic gradient descent. Left figure shows the learning rate transfers across depth in ResNets parameterized with Depth$-\mu$P. Right figure shows that for a fixed learning rate, the sharpness becomes depth independent during training. Parameters: batch size 64, epochs 10.

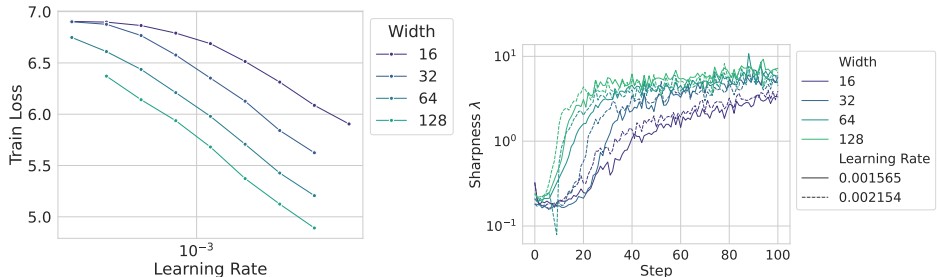

Figure 19: Residual convolutional networks (ResNets) trained on ImageNet with stochastic gradient descent. Left figure shows the learning rate transfers across width in ResNets parameterized with $\mu$P. Right figure shows that for a fixed learning rate, the sharpness becomes width independent during training. Parameters: batch size 128, epochs 1.

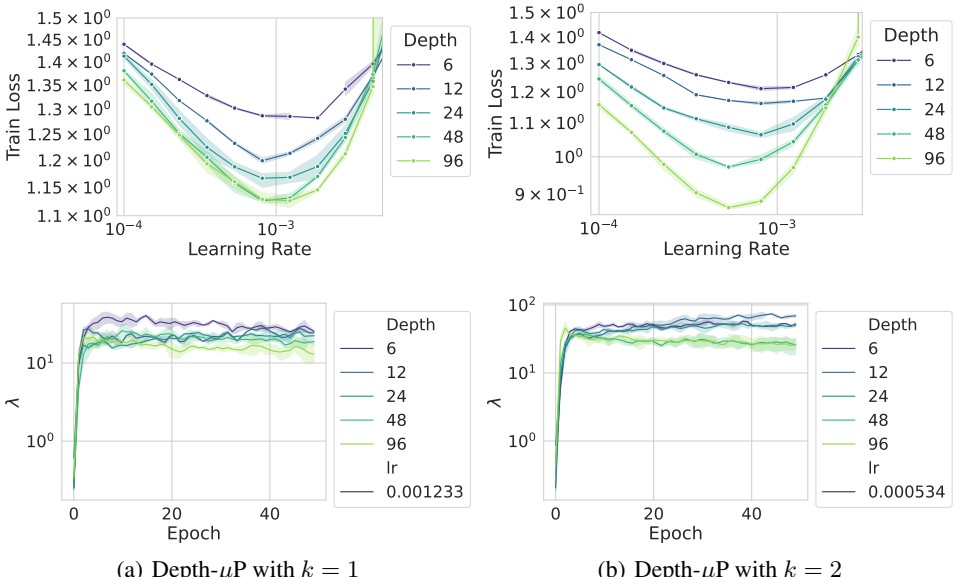

(a) Depth-$\mu$P with $k = 1$         (b) Depth-$\mu$P with $k = 2$

Figure 20: ConvNets trained with Adam with fixed learning rate on CIFAR-10. (a) One layer per block, showing that the learning rate transfers and we have sharpness Super Consistency. (b) When training with $k = 2$ layers per block, notice that the transfer worsens. While the effect is subtle in this setting, it translates to the sharpness accumulating finite-size effects during training. Here, we use the Depth-$\mu$P prescription for Adam reported in Table 1.

## G    Time Evolution of other Spectral Quantities

In this section we present convergence rates for various other spectral quantities of interest. In Figure 22, we show the same Super Consistency of the landscape in the case of Depth-$\mu$P, as exhibited in the width case in Figure 2. In Figure 23, we present the convergence rate of the largest NTK eigenvalue, sharpness and loss respectively, as well as the evolution of the Hessian trace during time. Figure 24 (a) shows the loss curve evolution under $\mu$P and NTP regime. Note in Figure 24 (b) and (c) how the sharpness achieves a near EoS value in the case of $\mu$P, whereas for NTP wider networks have a sharpness that remains closer to initialization. Finally, in Figure 25 we show that in the case of having multiple linear layers per block, this leads to violations of Super Consistency, and thus imperfect learning rate transfer. Note that this is the same case as in the GPT-2 plots illustrated in Figure 15.

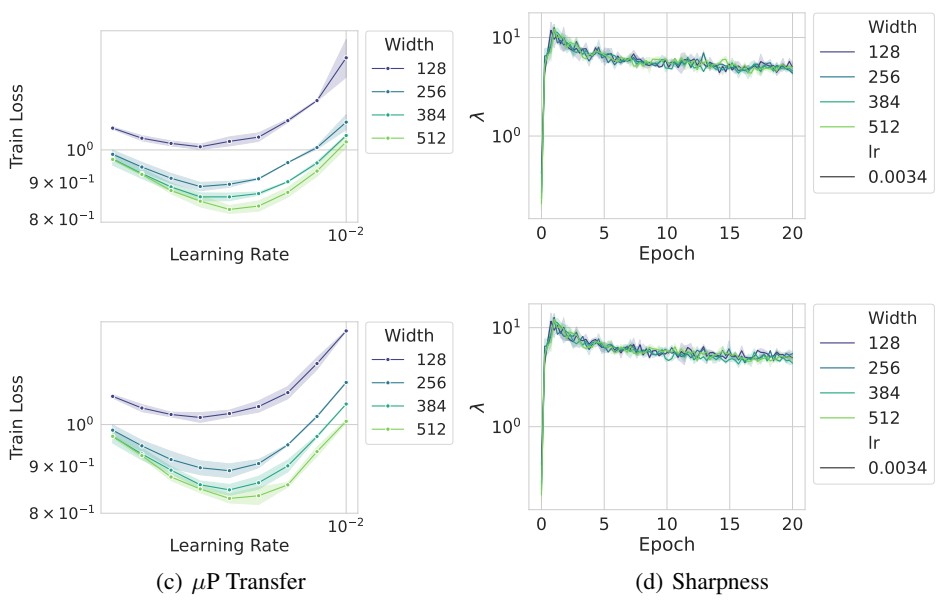

(c) $\mu$P Transfer          (d) Sharpness

Figure 21: Convolutional networks trained with Adam (top) and AdamW (bottom, weight decay 0.001) on CIFAR-10 parameterized with $\mu$P showing learning transfer in width (a) and super consistent sharpness evolution (b). HPs: 20 epochs, 3 layers, no skips, batch size 256, 200 warmup steps.

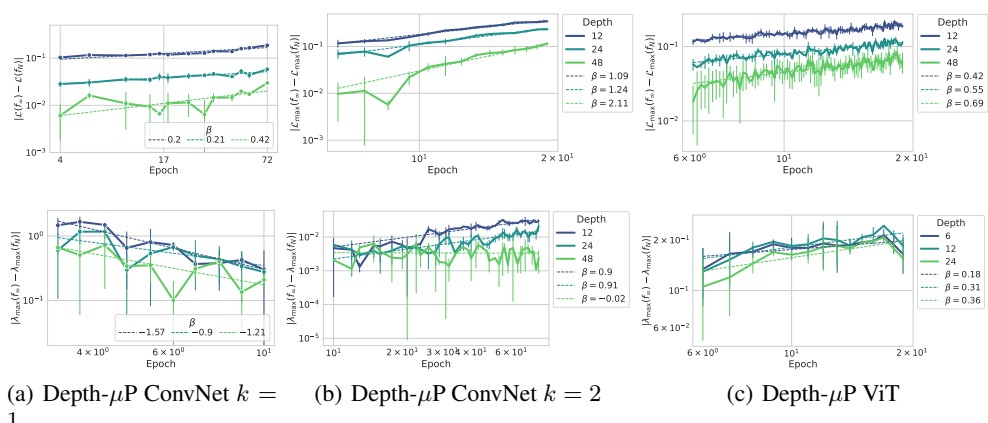

(a) Depth-$\mu$P ConvNet $k = 1$    (b) Depth-$\mu$P ConvNet $k = 2$      (c) Depth-$\mu$P ViT

Figure 22: Convergence rates with respect to time for the losses and sharpnesses shown in Figure 4.

## H Sharpness evolution in Random Feature Models

In this section, we compare the NTP and $\mu$P parameterizations to a random feature model, i.e. a model where we all the weights of the intermediate layers are frozen to their value at initialization, and only the final readout layer is trained. Crucially, this model does not learn features by construction for any learning rate and any width. The results are shown in Fig. 26. Notice how the transfer in the random feature model is achieved only at very large widths compared to $\mu$P. However, the transfer is better than in NTP. This is in line with our claim, as under a random feature model increasing the learning rate does not induce more feature learning, as is the case in NTP.

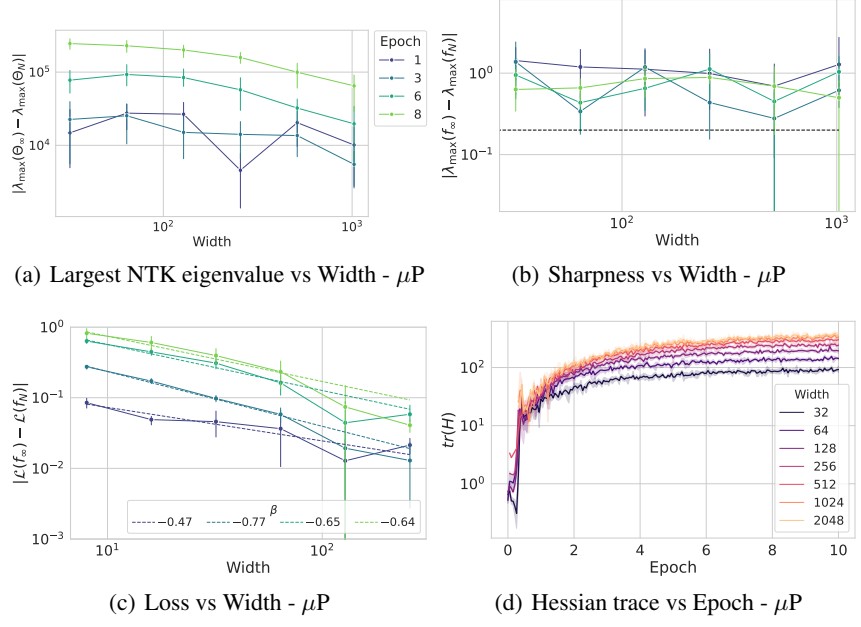

(a) Largest NTK eigenvalue vs Width - $\mu$P

(b) Sharpness vs Width - $\mu$P

(c) Loss vs Width - $\mu$P

(d) Hessian trace vs Epoch - $\mu$P

Figure 23: (a) Convergence rate of the largest NTK eigenvalue in width at multiple steps during training.(b) Convergence rate of the sharpness in width at multiple steps during training. Note that the largest NTK eigenvalue starts width independent for both learning rates, but becomes width dependent during training, as opposed to the sharpness which maintains a width independent dynamic throughout the whole optimization process (see Fig. 24). (c) Loss convergence rate in width; note that the loss accumulates finite-size effects during time, exhibiting a wider is better effect during training. (d) Hessian trace evolution during training at various widths. Unlike the sharpness, the trace has a width dependent period at the beginning of training, but approaches a width-indepedent threshold. Details: Residual ConvNet trained on CIFAR10 with cross-entropy loss.

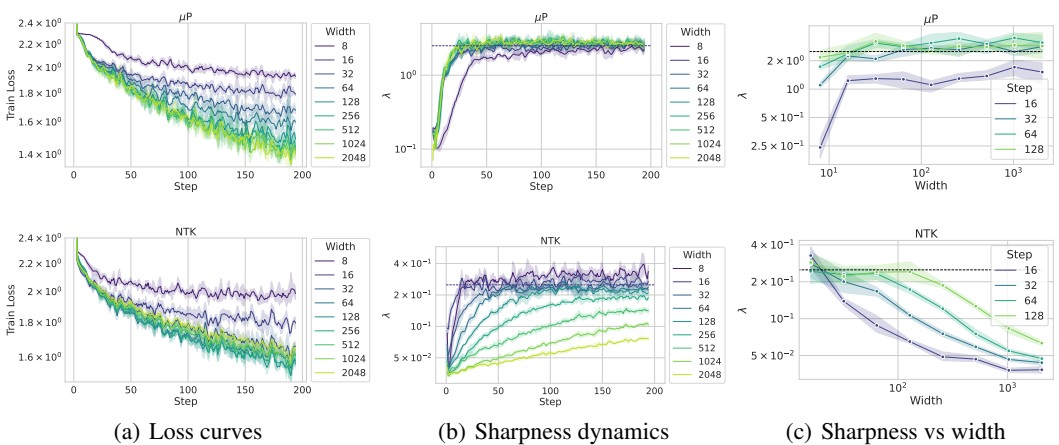

(a) Loss curves

(b) Sharpness dynamics

(c) Sharpness vs width

Figure 24: Early training dynamics in $\mu$P (top row) and NTP parameterization (bottom row). (a) Loss curves. (b): Sharpness dynamics. Notice how for $\mu$P progressive sharpening until EoS (black dashed line) is achieved at any width and with comparable speed, while for NTP the time to EoS progressively increase with width. Also, the loss curves start to depart from each other as training progresses, while $\lambda$ stays at EoS for a more sustained period ot time. (c) Sharpness vs width at selected time steps. For $\mu$P, $\lambda$ converges very fast in width, while in NTP it diminishes. Other parameters: architecture: Three-layer convolutional network. Dataset: CIFAR-10, without data augmentation. $B = 128$, epochs = 1, learning rate $\eta_0 = 0.8$ for $\mu$P and 8 for NTP. The reported width is the size of the readout layer.

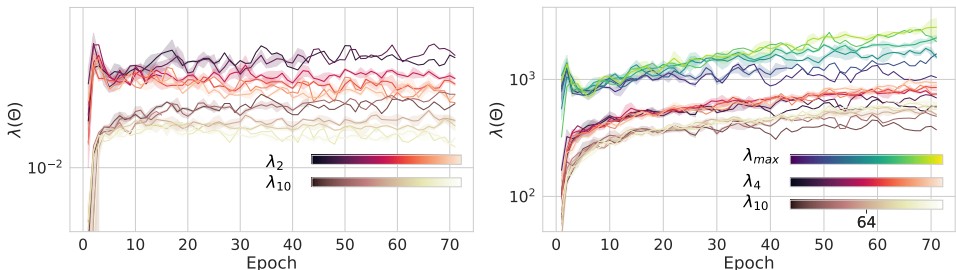

Figure 25: Dynamics of some of the eigenvalues of the Hessian (left) and NTK (right) under Depth-$\mu$P scaling with $k = 2$ layers per residual block. We observe violations on Super Consistency in both cases. The model is the same as in Fig. 4 (center). This is compatible with absence of learning rate transfer.

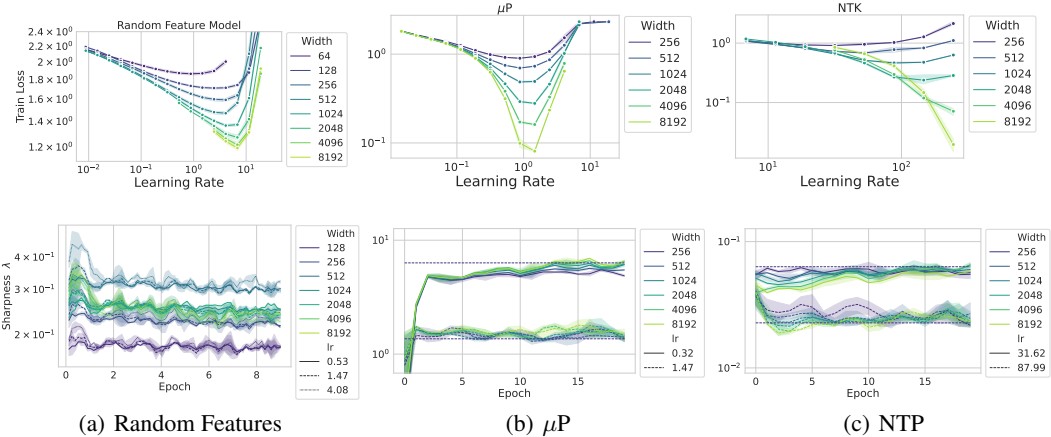

Figure 26: Learning rate transfer plot (top row) and sharpness dynamics (bottom row) for a three-layer convolutional network and three different settings. (a) random feature model (only the readout layer is trained), (b) and (c) correspond to $\mu$P and NTP parameterizations, respectively. In random feature learning, the absence of feature learning prevents the sharpness' evolution at any width, thus learning rate transfer coincides with the convergence of the sharpness $\lambda$ at initialization. Also notice how for NTP, the progressive sharpening converges to $\lambda = 2/\eta_0$ at a much lower speed as the width increases, in line with the early dynamics reported in Fig. 24. Other parameters: $B = 128$, epochs $= 20$ for the $\mu$P/NTP models and 10 for the random feature model, dataset: CIFAR-10, without data augmentation.

## I    Directional sharpness

In Figure 27 we provide the evolution of the directional sharpness, which captures the curvature along the gradient direction during training under $\mu$P and Depth-$\mu$P respectively in ConvNets. Note that, similar to the sharpness plots, the directional sharpness, defined as $\frac{\nabla_\theta \mathcal{L}^\top H \nabla_\theta \mathcal{L}}{\|\nabla_\theta \mathcal{L}\|^2}$ also follows a width independent trajectory during training. This measure has been used, for instance, in Gur-Ari et al. [17].

## J    Experimental details

The experiments were ran on A100 and H100 GPUs, with $80$GB VRAM. Each experiment averaged less than 24 hours total execution time. Unless stated otherwise, we use data augmentations, where the random transformations are compositions of crops, horizontal flips, and 10-degree rotations. Additionally, we provide further details on the models used in our experiments and the modifications we have introduced.

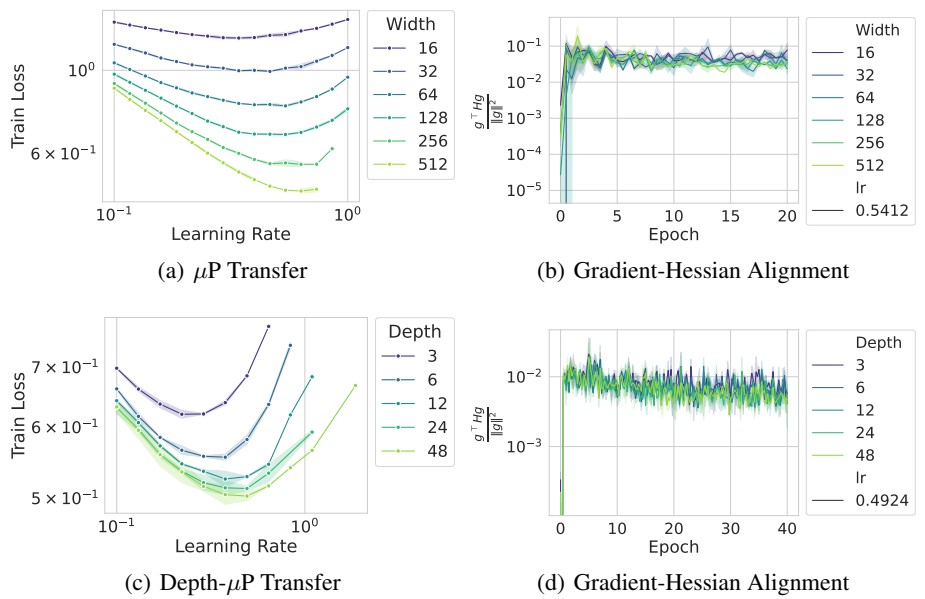

(a) $\mu$P Transfer         (b) Gradient-Hessian Alignment

(c) Depth-$\mu$P Transfer         (d) Gradient-Hessian Alignment

Figure 27: Convolutional networks trained with SGD on CIFAR-10 parameterized with $\mu$P (top) and Depth-$\mu$P (bottom) showing learning transfer (a) and super consistent Hessian-gradient alignment during training (b, d). HPs: (top) 100 warmup steps, batch size 256, 20 epochs, no residual connections (bottom) 200 warmup steps, batch size 128, 40 epochs, $\frac{1}{\sqrt{L}}$ scaling (both) 6 layers, ReLU.

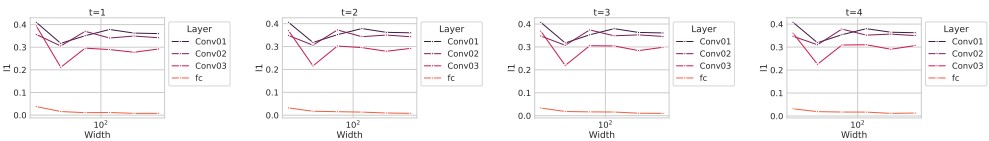

Figure 28: Coordinate check in $\mu$P.

## J.1 Hessian Computation

The implementations of our models are done in PyTorch. For the Hessian measurements, we use PyHessian [94]. In particular, the library adopts the Power Iteration method for the top $k$ Hessian eigenvalues (i.e. including the sharpness) and the Hutchinson method for trace computation. Both methods adopts Hessian vector products to avoid computing the whole Hessian. This reduces the time complexity from quadratic to linear in the number of parameters. In both algorithms, we fix the number of iterations and tolerance between consecutive eigenvalue computation to the default values of 100 and 0.001, respectively. We measure the sharpness on the same fixed batch throughout training.

**SGD** Among the equivalent ways of parametrizing then network with $\mu$P, we opt for the one that rescales the learning rate by the width, i.e. $\eta = \eta_0 \gamma^2 = \eta_0 N$. This effectively sets the EoS threshold to the width-dependent value of $2/(N\eta_0)$. In our plot, we take this scaling difference into account by computing the eigenvalues of the scaled Hessian $\gamma^2 H = NH$. With respect to learning rate transfer, such a rescaling makes intuitive sense, as it is $\eta_0$ that is transferring, and not $\eta$.

**Adam** Adam updates are of the form $\theta_{t+1} = \theta_t - \eta P_{t+1}^{-1} m_{t+1}$, where $m_t$ is a momentum vector computed in the exponential moving average fashion: $m_{t+1} = \beta_1 m_t + (1 - \beta_1) g_{t+1}$, where $g_t$ are the gradient of the loss at time $t$. $P_{t+1}$ is a diagonal preconditioner of the form

$$P_t = (1 - \beta_1^t) \left[ \text{diag}\left( \sqrt{\frac{\nu_t}{1 - \beta_2^t}} \right) + \epsilon I \right],$$

where $\nu_t := \beta_2 \nu_{t-1} + (1-\beta_2)g_t^2$, where $g_t^2$ is the element-wise squared gradients. In the experiments of this paper, we consider the preconditioned Hessian $P^{-1}H$, for which it is shown that its largest eigenvalue converges to the EoS threshold of $\frac{2(1+\beta_1)}{\eta(1-\beta_1)}$ in Cohen et al. [61]. Furthermore, we use the Adam $\mu$P parametrization in Table 8 of Yang et al. [6], which rescales the learning rates of the hidden layers (i.e. with both input and output dimensions that scale with $N$) by $1/N$. To account for this, we further adjust the Hessian by computing $\tilde{H} = DP^{-1}H$, where $D$ is a diagonal matrix containing the learning rate for each parameter. We always report spectral quantities of $\tilde{H}$ in the experiments with Adam. With these modifications, we expect $\lambda_{\max}(\tilde{H}) = \frac{2(1+\beta_1)}{(1-\beta_1)}$, which is 38 for the full-batch case and the default $\beta_1 = 0.9$. Indeed, this is what we observe in our experiments. We stress that we expect Super Consistency of this preconditioned Hessian, where in $\mu$P different layers may have a different dependence on the width [6]. Finally, we point out that in the Depth-$\mu$P experiments, we reported the hessian of $N/\sqrt{L}H$. Although this produces results that deviate from the prescription of Cohen et al. [61], it is sufficient to capture Super Consistency in depth, as each layer has the same depth-dependence.

## J.2 GPT-2

The backbone architecture in the experiments presented on Wikitext is the standard GPT-2 transformer introduced by [33], with the Depth-$\mu$P parameterization changes presented in [6, 8, 53]. Crucially, the following modifications are introduced by the $\mu$P parameterization:

- The attention map is rescaled by $\frac{1}{d_Q}$, as opposed to $\frac{1}{\sqrt{d_Q}}$

- The residual branch is downscaled by $\frac{1}{\sqrt{L}}$, where $L$ is the number of layers in the model

Our implementation is based on the implementation provided by Yang et al. [6], with the addition of the residual scaling. This uses a different parametrization from the one reported in Table. 1 but equivalent dynamics, obtainable using their "abc-rule". Similar to the experiments performed on ViTs, the GPT-2 models are trained using Adam, where the base width is fixed and the depth is varied (and vice versa for the Depth-$\mu$P case). In addition, for the fixed width, increasing depth experiments, we place the layer normalization layer in front of the residual, following the Pre-LN architecture, unless stated otherwise, and we do not use any learning rate warmup. Note that in this setting we also train without the $1\sqrt{L}$ scaling of the learning rate that the theory would prescribe (summarized in Table 1). This follows following the heuristic prescription of Bordelon et al. [7]. As in their case, we empirically observed that we did not get learning rate transfer if we used the scaling. In the fixed depth, increasing width case, we use a linear rate warmup, with no decay, and we use a standard Post-LN GPT-2 style architecture.

## J.3 Vision Transformers (ViTs)

The ViT implementation is based on the work of [95] and follows the same tokenization and training protocol. In order to follow the Depth-$\mu$P parameterization, we make the same modifications as in J.2. The models are trained with Adam.

## J.4 ResNet

We use convolutional neural networks with skip connections, with $3 \times 3$ kernels and stride 1. We apply pooling after every 3rd layer, followed by a subsequent convolutional layer. Following Bordelon et al. [7] and Yang et al. [8], we downscale the residual branches by $1/\sqrt{L}$. Note that in the Depth-$\mu$P setting we also scale the learning rate as $1\sqrt{L}$ in these experiments.

## J.5 Coordinate check for $\mu$P

In Figure 28 we check the coordinatewise evolution of the activations at hidden layers within a ConvNet. Note that the evolution is width independent, as predicted by $\mu$P theory.

# K   Summary of Feature Learning Parametrizations

Here we summarize the parametrizations used. In Table 1, we report the scaling of the learning rate, output multiplier $\gamma$, depth-scaling exponents $\alpha$ for the residual branches for Depth-$\mu$P (both SGD and Adam) [7, 8, 96]. $\mu$P is recovered as a special case, where the depth dependence is ignored. Notice that this version of Depth-$\mu$P for Adam is obtained with a simplification, setting Adam's $\epsilon$ parameter to zero, where Sign-GD is recovered.

|      | $\eta$ | $\gamma$ | $\alpha$ | Non residual block layers |
|------|--------|----------|----------|---------------------------|
| SGD  | $\eta_0 \gamma^2$ | $\gamma_0 N^{\frac{1}{2}}$ | $\frac{1}{2}$ | 1 |
| Adam | $\eta_0 N^{-\frac{1}{2}} L^{-\frac{1}{2}}$ | $\gamma_0 N^{\frac{1}{2}}$ | $\frac{1}{2}$ | $L^{\frac{1}{2}}$ |

Table 1: Summary of Parametrizations. The non residual block layers represent those trainable vectors/matrices that are not in a residual block (typically, the first and last ones). For these layers, we prescribe how to rescale both the weight variance and scaling multiplier.

