# OpenReview forum: "Super Consistency of Neural Network Landscapes and Learning Rate Transfer"
_NeurIPS.cc/2024/Conference — NeurIPS 2024 poster_

### Official Review · Reviewer_ssWb · 2024-07-10

**Soundness:** 4
**Presentation:** 4
**Contribution:** 4
**Rating:** 8
**Confidence:** 2

**Summary:**

The paper investigates the loss landscape of the model with scaling width or depth,  through observing the largest eigenvalue of the Hessian marix and the NTK matrix. Authors show empirically that the loss Hessian evolve almost identically for different model sizes (which is named Super consistency), however, ntk accumulates finite-size effects over time. Authors also validate their empirical findings using theory in a two-layer NN with linear activations.

**Strengths:**

- The paper discusses an interesting phenomenon that the loss landscape gradually becomes stable, this might explain the transfer of learning rate
- The paper shows NTK (lazy learning) has different behavior when comparing with Hessian of the loss, this suggests NTK is insufficient to explain the behavior
- The paper gives an explicit evolution law under 2-layer linear NN setting,

**Weaknesses:**

- The authors do not conduct larger-scale environment since needing to compute Hessian

**Questions:**

- I am confused about Line 209, this line mentions we can decompose H into G + R, while G = K^T K and the dynamics are mainly driven by G. NTK and G share the same nonzero eigenvalues, doesn't this mean H and NTK should have similar behavior? Or feature learning happens in H which makes the difference between NTK and Hessian?

**Limitations:**

The authors have fully addressed the limitations in their paper

---

> ### Author Rebuttal · Authors · 2024-08-07
>
> We thank the Reviewer for the strong overall score and for marking “excellent” our paper across all the three evaluation axes.
>
> On the weakness:
>
> 1. **Larger scale experiment**: We managed to scale the Hessian computation up to 300 million parameters in a Transformer model trained on Wikitext with Adam (width scaling), surpassing the scale reached in the reference literature on the EoS (Cohen et al, 2021). Please consult Figure 1 in the one-page pdf attached. We find that Super Consistency still holds at this scale. See also the global response on this point.
>
> Questions:
> 1. **Hessian vs NTK dynamics**. Under $\mu$P, both the Hessian and NTK evolve from initialization, a distinctive feature of feature learning scaling limits. These two quantities are indeed related by the Gauss-Newton decomposition of the Hessian, which for MSE loss reads $H = K^T K + R$. Indeed, for MSE loss the dynamics of the NTK and Hessian are connected, in that the change of the Hessian is largely driven by the change in the NTK (Figure 5). However, notice that the matrix $R$ still plays a role and cannot be neglected when the network is far from convergence (more subtly, the interaction between K and R gives the self-stabilization property that preserves edge of stability for $\lambda_{\max}$). Secondly, under cross-entropy loss, the first term of the decomposition is not $K^T K$ but $K^T D K$, where $D$ contains the second derivative of the loss for different datapoints. Thus, there is an additional term that potentially makes the dynamics of the Hessian different from the NTK. Indeed, this is what we observe in Figure 2, where the NTK eigenvalues accumulate significant finite-size effects, while the Hessian top eigenvalues are Super Consistent.
>
> We hope that the new experiments and clarification will make the Reviewer more confident about their evaluation.

---

> > ### Comment · Reviewer_ssWb · 2024-08-09
> >
> > Thank you for your detailed response!

---

### Official Review · Reviewer_47kG · 2024-07-10

**Soundness:** 3
**Presentation:** 3
**Contribution:** 3
**Rating:** 7
**Confidence:** 3

**Summary:**

The authors argue that the top eigenvalues of the loss Hessian stabilize throughout training under width and depth muP scaling. This phenomenon is called the Super Consistency of the loss landscape. The authors provide theoretical convergence guarantees and empirical experiments supporting their claims. The learning rate transfer under muP scaling also correlates with the super consistent landscape, i.e. optimization follows the trajectories of sharpness (top eigenvalue of the loss Hessian). Under NTK or other suboptimal scaling, the super consistency is also violated and thus learning rate transfer failed. The authors further show that the dynamics of the weights are fully specified by a set of governing equations and thus one can derive a theoretical edge of stability result under scaling. The phenomenon of progressive sharpness towards stability also happens along with the NTK evolution with finite-size effects, suggesting other factors contributing to the super consistency of the sharpness.

**[raising score from 6 to 7 after rebuttal]**

**Strengths:**

1. The observation about the super consistency of the loss landscape and its relation to learning rate transfer under muP scaling is quite an important contribution to the community. I’m not an expert in this field, but this result seems novel to me.
2. The paper is well-written and easy to follow.

**Weaknesses:**

The empirical evaluations are a bit limited.

- First, a lot of the claims are made under the setting of ConvNet on CIFAR-10. It’s understandable from a compute perspective, but it’s not clear if super consistency scales to even larger models.
- Second, the GPT-2 results on WikiText seem to break the super consistency, though it could be possible that it’s due to the Transformer block itself as the authors explained. So it’s not clear if the super consistency results hold across different models and modalities.

**Questions:**

1. How are you getting the eigenspectrum of the loss Hessian? It seems possible to get at least a few top eigenvalues using iterative Lanczos methods for models larger than 124 million parameters GPT-2 with A100-80GB.
2. I don’t find enough evidence that the super consistent sharpness causally helps the learning rate transfer under muP scaling. In line 178, “optimization happens along trajectories of super consistent sharpness λ_max”. Is there evidence that the optimization does happen first along this direction, i.e. some kind of spectral bias during learning?

**Limitations:**

See weakness and questions

---

> ### Author Rebuttal · Authors · 2024-08-07
>
> We thank the reviewer for acknowledging the importance and novelty of our results. Here we address the concerns:
>
> 1. **A more diverse set of experiments**: we would like to gently push back on this point. We have already performed experiments on CIFAR-10, Imagenet (vision), and wikitext (language) with two different architectures: ResNets and Transformers (including Vision Transformers). We have also performed experiments on several parametrizations, including NTK, $\mu$P and SP in width, and Depth-$\mu$P and residual networks without $1\sqrt{L}$ scaling for depth. We have trained models with up to 96 layers in depth (e.g. Figure 19), and 8172 units in width. We have other interesting ablations, including batch size, data augmentation, and different optimizers (SGD, Adam, now including AdamW as well, see the one page pdf included in the global response).
>
> 2. **New Evidence on Transformers at scale and convolutional networks**. We have performed several new experiments on *width* scaling of Transformers (with Adam) and convolutional networks (with Adam and AdamW). In contrast to depth-scaling, where Super Consistency was not great due to the nature of the Transformer layer, in this case we observe Super Consistency to hold from very small to very large width Transformers’ (scaling up to about 300 million parameters for the largest model) and the learning rate to transfer significantly better. Please see Figures 1 and 2 of the one-page pdf. In particular, our results are based on Post-LN (like GPT-2) models trained on wikitext (with Adam) and Covnets on CIFAR-10. We hope that this extra experimental evidence addresses the Reviewer's concerns about the presence of Super Consistency across more models and datasets.
>
> 3. **Absence of Super Consistency in Transformer experiment (with $\mu$P-Depth)**: as the Reviewer mentions, we do hypothesize that the observed breach of super consistency is due to the nature of the Transformer block design, having multiple layers per residual block. This breach is thus expected, and we have also reproduced it in the ResNet case with 2 layers per residual block, as in Figures 4, 19 (a,b), and 20 (a,b,c). Furthermore, the absence of Super Consistency correlates with worse learning rate transfer, and it is thus compatible with our claims.
>
> Questions:
>
> 1. **Hessian Computation**. We use PyHessian to compute the eigenvalues, which adopts the Power iteration method to get the eigenvalue spectrum. We use a large fixed batch of 1024 datapoints for Hessian estimation. Models with more than 124 million parameters would certainly fit a single A100 GPU, but the estimation becomes very slow and thus cannot be performed at the same frequency as a smaller model. However, as shown in Figure 1 of the one-page pdf, we have run a Post-LN Transformer with up to about 300 million parameters and observed that super consistency still holds at this larger scale.
>
> 2. **SGD along Hessian directions and learning rate transfer** Indeed, there is compelling evidence (Gur-Ari et al, 2018) that SGD happens in the subspace spanned by the top $k$ Hessian eigenvectors, where k is the number of classes. In Figure 2 of the paper, we report how a subset of the top 10 eigenvalues behaves across time and model scale. In most of the other experiments of the paper, we restrict to top eigenvalue $\lambda_{\max}$ due to computational reasons and due to its importance in learning rate choice. Please see Appendix A for a thorough discussion on the importance of the sharpness in the choice of the learning rate, before and after the EoS era. Regarding the specific wording, by that sentence, we meant that “lambda_{\max} is super consistent across the training trajectory”. We made this clear in the revised version of the paper.
>     + New Evidence on Gradient-Hessian alignemnt. We performed an experiment where we computed $g^THg/||g||^2$, where g are the weight gradients and H is the loss Hessian, thus taking into account the curvature along the gradient direction. The results are in Figure 3. We observe that Super Consistency largely holds in this case as well.
> Finally, please notice that all our experiments show a clear correlation between sharpness super consistency (in general, top k eigenspectrum) and learning rate transfer. Due to the importance of the sharpness in determining the trainability of the model at large learning rates, our results provide strong empirical evidence for a potential explanation of the phenomenon of learning rate transfer. However, more work has to be done in neural network optimization to establish (i.e. prove) how the optimal learning rate depends on the Hessian spectrum (together with other quantities).

---

> > ### Comment · Reviewer_47kG · 2024-08-09
> >
> > Thank you for the additional experiments and clarifications. I'm raising my score to 7, though I still think the following two points can be addressed further to improve the paper, but it could be too demanding in the short time frame of the rebuttal:
> >
> > 1. empirically show that super consistency scales to larger model. This is possible via parallel model sharding across more GPUs from the system level. It's also possible to use more efficient numerical algorithms other than power iteration. I'm curious if super consistency holds or even improves as a function of model scale.
> >
> > 2. For the point "more work has to be done in neural network optimization to establish (i.e. prove) how the optimal learning rate depends on the Hessian spectrum (together with other quantities)", I believe this is worth investigating further, but I know I am asking for too much for the scope of this paper.

---

> > > ### Author Response · Authors · 2024-08-12
> > >
> > > We would like to thank the reviewer for the points they brought up, as well as for the score increase. Regarding the additional points:
> > >
> > > 1) We agree that it would be interesting to see how Super Consistency behaves at larger scales. In the final version of the paper, we will aim to scale up models further using the suggestions proposed by the reviewer.
> > >
> > > 2) We also believe that a very interesting future avenue would be to theoretically study the relationship between the optimal learning rate and the Hessian eigenvalue spectra. Understanding this dynamic could lead to better schedulers or even optimization algorithms that could speed up convergence in large models.

---

### Official Review · Reviewer_11Ly · 2024-07-12

**Soundness:** 3
**Presentation:** 3
**Contribution:** 3
**Rating:** 6
**Confidence:** 4

**Summary:**

This paper proposes the concept of super consistency, which describes the stable properties of the loss landscape during training. By analyzing the maximum eigenvalue of the Hessian matrix, it is found that the sharpness under the μP and Depth-μP frameworks remains super-consistent and stable near the threshold. In contrast, NTK and other frameworks show significant differences.

**Strengths:**

This paper proposes a new concept of "super consistency", which provides a new perspective for understanding the behavior of models of different scales. Through a large number of experiments, the learning rate migration phenomenon under the μP and Depth-μP frameworks is verified, and the consistency of these frameworks on different tasks and datasets is demonstrated, including ResNets, Vision Transformers, and GPT-2.

**Weaknesses:**

This paper provides some theoretical analysis, mainly focusing on two-layer linear networks and failing to fully verify the theoretical applicability in nonlinear networks and complex structures.

**Questions:**

Is it feasible to extend the results of this article to other algorithms, such as AdamW?

**Limitations:**

The theoretical setting i simple. The author can consider trying to expand the two-layer linear network to two-layer ReLU or deep linear network.

---

> ### Author Rebuttal · Authors · 2024-08-07
>
> We thank the Reviewer for the careful assessment of our paper, and for highlighting the extensive suite experiments that we run, including different scaling regimes ($\mu$P, Depth $\mu$P, SP, NTK, etc..) and architectures. On the weaknesses:
>
> 1. **Theoretical limitation**: please notice that our paper is the first that analyzes a two-layer network in this generality. The closest work to our paper is Kalra et al, 2023. In contrast to this work we allow multiple datapoints instead of a single data point study, learning targets different than 0. Additionally, we use the standard definition of sharpness (largest eigenvalue of the loss Hessian) instead of the Hessian trace as a proxy. Extending our framework to nonlinear (and deeper) networks is certainly a fascinating avenue for future work. However, it will likely incur significant difficulties in computing the sharpness of deep nonlinear networks during training. Finally, please notice that our paper is mainly empirical, and the scope of the theory is mainly to provide intuition and justification of the empirical results. Under this perspective, the two-layer linear network is an interesting tractable model that expresses the phenomenology observed in practice for more complex networks.
>
> 2. **AdamW**: We have performed new width-scaling experiments with Adam and AdamW on 3 layers Convolutional Networks, extending Super Consistency to these settings. The weight decay scaling in AdamW follows the settings of Wang, X. and Aitchison, L., 2024 and Yang et al., 2022. Note that in the added experiments on Convolutional Networks, the learning transfers across width, and we have a Super Consistent sharpness evolution during training. We also believe that our results could be empirically extended to other optimization algorithms as well. See Figure 2 of the one-page pdf for our training runs on CIFAR-10.  We note that a very interesting future work avenue would involve understanding the role of adaptivity in Super Consistency from a theoretical point of view.

---

> > ### Comment · Reviewer_11Ly · 2024-08-11
> > **Response**
> >
> > Thank you for the additional experiments and clarifications. Your response has addressed my concerns.

---

### Official Review · Reviewer_f2zq · 2024-07-12

**Soundness:** 2
**Presentation:** 1
**Contribution:** 3
**Rating:** 7
**Confidence:** 5

**Summary:**

The authors conduct a series of experiments in which they investigate for which attributes of the neural network the gap between its infinite-width (or infinite depth) value and the finite counterpart grows or shrinks during training. Among other attributes, they look at the training loss, largest loss Hessian eigenvalue, and largest NTK eigenvalue. They investigate how the results for different parameterisations (such as muP, neural tangent parameterisation, and some other unstable parameterisations with no well-defined infinite width behaviour).

**Strengths:**

The asymptotic scale properties of neural network training are an important area of research. This paper investigates an interesting question: how do the properties of the loss landscape, such as the curvature along the training trajectory, evolve as one scales up the neural network size.

Furthermore, the experimental results are interesting, and well-presented in the figures. I liked Figure 3 in particular, which most convincingly illustrates the claim that the gap between finite-width and infinite-width attribute value shrinks/grows throughout training. In fact, I wish for every hypothesis investigated, a plot like that in Figure 3 was shown.

Lastly, it appears the authors did proper ablations, investigating multiple datasets and models, to verify their empirical conclusions.

**Weaknesses:**

The paper falls short in presentation and the formalism.

One of the largest issues is the definition of the term “super consistency”. Definition 3.1 has several issues:
- The authors defines $S_N(t)$ as a function of the predictor $f_t(x)$. This is pretty vague, and not sure matches what authors are trying to do. If the predictor $f_t$ is interpreted to be a function $f_t:\mathcal{X}\to\mathcal{Y}$ from some input space to some outputs space implied by the neural network architecture and weights at time $t$, then $S_N(t)$ cannot capture something like the curvature of the loss with respect to the weights. I'm convinced this hurdle can be overcome by carefully defining all objects, and what they are (e.g. a ‘predictor’).
- If $S_N(t)$ depends on the weights (such as when considering the spectral norm of the loss Hessian), then it's a random variable. Hence, all the notions of distance and convergence need to be defined for random variables for the expressions in Definition 3.1 to make sense.
- It's not clear the limit $S_\infty(t)=\lim_{N\to\infty}S_N(t)$ exists for many properties being considered. In fact, this already precludes the authors from talking about super-consistency of parameterisations that do not have well-defined infinite-width limit training dynamics (e.g. SP).
- line 127 “if at any finite $N\geq N_0$:” – did the authors mean to say ‘there exists some $N_0$ such that for all $N\geq N_0$’? Otherwise this sentence doesn't make sense.
- Does the symbol $\sim$ here represent asymptotic equivalence as $t\to\infty$? This is something that should have been defined.
	- “$\sim$ denotes the finite-time behaviour during training” is not a formal definition. I have no idea what it's meant to say.
	- Also, the whole condition of “$|S_N(t)-S_\infty(t)|\sim g(t)$ where $g(t)$ is a non-increasing function of time...” seems like it could have been equivalently stated much more simply as $|S_N(t)-S_\infty(t)|=\mathcal{O}(1)$ as $t\to\infty$.
- This definition seems different from how super consistency was described in the abstract introduction: “certain [...] properties [... are largely independent] of the width and depth[...].
	- We name this property super consistency of the landscape”. The concept of super consistency, as defined in Definition 3.1, is simple enough to express in two sentences, that I don't see a reason it can't be explained in the introduction and/or abstract properly.

Later in the paper, the authors proceed to use ‘super consistency’ in a sense completely disjoint from that in Definition 3.1. On lines 168-159 they say: “The optimal learning rate is preserved across widths/depths, indicating very fast convergence with respect to the scaling quantity (i.e. it's super consistency).” Definition 3.1. has seemingly nothing to do with the speed of convergence in the scaling quantity (width/depth).

In fact, I think the paper would have been stronger had the authors cut back on formalising things like “super consistency”, and just presented the empirical results for what they are. The term “super consistency” doesn't strictly seem necessary to convey the take-aways of the paper, and, at the moment, makes the paper more convoluted. Of course, a thorough formalism might be preferred, but it its current form it detracts from the paper.

Others:
- I find the phrase “optimization happens along trajectories of super consistent top Hessian eigenvalues” quite confusing. It took me a while to realise the authors are saying that the ‘Hessian eigenvalues are super consistent along the optimisation trajectory’, which is the way I'd recommend they phrase it throughout.
- There are several typos throughout the paper. Sometimes, seemingly parts of mathematical expressions are missing (e.g. line 161).

**Questions:**

- Why are there two sets of lines for each width on Figure 1, one solid one dashed?
- Is the Appendix G, which is meant to discuss “the effect of the lower order eigenvalues”, empty?
- In Figure 2, the authors look at 4-th and 10-th largest eigenvalue. How come the authors decided to look at a fixed N-th largest eigenvalue? Given the scaling in width, wouldn't it be more interesting too look at say 4th and 10th percentile largest eigenvalue, given that the number of eigenvalues grows with depth/width?

**Limitations:**

- Definition 3.1 concerns asymptotic properties in *both* width/depth and training time, which are then doubly difficult to establish with certainty with finite width, depth, and training time experiments.

---

> ### Author Rebuttal · Authors · 2024-08-07
>
> We sincerely thank the reviewer for the detailed feedback and for the further discussion that we anticipate. Also, we thank the reviewer for acknowledging certain strengths of the paper, such as the importance and validity of our findings in the context of understanding neural networks’ loss landscape at different scales.
>
> The reviewer’s main concern is regarding how Super Consistency is defined (Definition 3.1). The main purpose of putting forward the definition was to have an actionable measure of Super Consistency that would reflect the precise quantitative results of Figure 3 and it is not supposed to be used in a mathematical theory. This seems to be in line with the Reviewer’s suggestions to “cut back on formalising things like “super consistency”. However, we do agree with the Reviewer that certain aspects of the presentation may seem to introduce mathematical formalism, which is not intended due to the experimental nature of the presented results. Thus, we decided to describe superconsistency outside of a formal definition environment, taking into account the Reviewer's valuable feedback. We hope that the Reviewer agrees that this is a more sensible choice in the context of our paper.
>
> **To clarify the reviewer’s confusion on the concept of Super Consistency**: we started from the concept of Consistency in Vyas et al (2023) which refers to how certain aspects of the network (such as the logits and the loss function) are the same at different scales *early in training*. We extend this concept to Super Consistency by requiring that Consistency is maintained for a longer period of training time, as we state in the abstract (lines 8-9) and introduction (lines 34-36). In the revised version of the paper, we try to achieve this objective once again, hoping to resolve the confusion. For your reference, here is the new phrasing of Section 3, replacing Definition 3.1:
>
> *Super Consistency refers to when certain aspects of the loss landscape and of the predictor $S_N(t)$ (in this paper $S_N(t)$ refers to the NTK's and loss Hessian's eigenvalues or the loss itself) exhibit the following two properties:*
>
>  *1. At large $N$, $S_N(t)$ does not deviate significantly from its limit $S_{\infty}(t) := \lim_{N\to\infty}S_N(t)$.  This is what is referred to as consistency in Vyas et al, 2023.*
>
>  *2. $S_N(t)$ does not accumulate significant finite size effects over time, i.e. the curves of $S_N(t)$ and $S_\infty(t)$ remain close over a sustained period of training.*
>
> *With respect to the experiment illustrated in Fig. 1, notice that the curves of the loss (center) at different widths show progressive and significant deviations, thus violating Super Consistency. On the other hand, the sharpness dynamics for $\mu$P qualitatively exhibit little-to-no deviations. Also, notice that we assume the existence of the limit $\lim_{N\to\infty}S_N(t)$. For those parametrizations (e.g. standard parametrization) that do not have a well-defined limit, $S_N(t)$ diverges at large $N$ and Super Consistency is trivially violated.*
>
> And a few lines later:
>
> *To give a quantitative measure to the finite-size accumulation property, we measure deviations over time by estimating the following quantity:
> $$g(t) := |S_{N}(t) - S_{\infty}(t)|.$$
> When $g(t)$ is an increasing function (up to fluctuations), Super Consistency is violated.*
>
> The current phrasing simplifies the notation (addressing the first 5 points in the weakness Section), and clarifies the intended meaning of Super Consistency in the abstract and intro (addressing the 6th point).
> We hope that the clarifications on the goal of the definition, together with the new simplified version of the corresponding Section solve the Reviewer’s concerns about the formalization of the Definition and its underlying meaning.
>
>
> Others:
> 1. “I find the phrase “optimization happens along trajectories of super consistent top Hessian eigenvalues” quite confusing”. This wording was intended and comes from a bias on related literature, such as “Optimization happens at the edge of stability” or “Gradient Descent Happens in a Tiny Subspace”. However, we agree that it might create slight confusion and have changed it according to the Reviewer’s recommendation.
>
> 2. The typo in line 161 has been fixed.
>
>
> *Questions:*
>
> 1. **Why are there two sets of lines for each width in Figure 1, one solid and one dashed?** Referring to the left column, different line styles indicate different learning rates, we plot 3 learning rates for $\mu$P and 2 learning rates for NTK to not make the plots too cluttered.
>
> 2. **Appendix G**. We thank the reviewer for pointing this out. We have now completed the section.
>
> 3. **How come the authors decided to look at a fixed N-th largest eigenvalue?** The reason for this is that earlier work established that SGD dynamics largely happens in the subspace spanned by the top $k$ Hessian’s eigenvectors, where $k$ is the number of classes. In our setting, $k$ is a constant independent of the width. Also, due to EoS-type of results, the largest eigenvalue would be converging to $2/\eta$ regardless of the width. However, in the direction proposed by the Reviewer, we have performed a new experiment where we measure the curvature along the gradient direction, which is a more global measure in the sense that it takes into account the whole Hessian. Super Consistency still holds (See Figure 3 of the one-page pdf)
>
> **Other remarks:**
>
> 1. **More figures like Figure 3.** The Reviewer appreciated the experiments on the distance between finite and infinite models over time as in Figure 3, suggesting that more plots like that should be included for other experiments. Please notice that Figures 20 and 22 also show this kind of analysis for other experiments.
>
> 2. **On the time horizon.** We operate at large training time, but not to complete convergence, as we show in Appendix E.1), thus we avoided talking about time asymptotics of the sharpness dynamics.

---

> ### Author Response · Authors · 2024-08-07
> **Rebuttal by Authors**
>
> Finally, we thank the reviewer for their valuable and precise feedback, and we would like to gently push back on the score of 1 given to the paper presentation. The general consensus amongst the other reviewers is that the main body of work is well-written and well-presented and we would kindly ask the reviewer to reconsider their score.

---

> > ### Comment · Reviewer_f2zq · 2024-08-09
> > **Response 1**
> >
> > Thank you for engaging on the points regarding presentation, and for trying to work with the feedback in the review to improve it.
> >
> > I'll try and go through the author's rebuttals, and point out where I think issues still remain.
> >
> > > Super Consistency refers to when certain aspects of the loss landscape and of the predictor $S_N(t)$ (in this paper $S_N(t)$ refers to the NTK's and loss Hessian's eigenvalues or the loss itself) [...]
> >
> > I think that is an improvement.
> >
> >
> > I'm familiar with the (Vyas et al. 2023) work. As far as I know, they don't have a formal or semi-formal definition of consistency, but my understanding is it just colloquially means that some quantity is close to the infinite-width counterpart, whenever that limit is well-defined (e.g. muP, NTP). The key interesting part of their paper is not that some quantities become consistent (that directly follows from what it means for a limit to exist), but that they do so at realistic widths and depths.
> >
> > I think the definition you ascribe to the term consistency:
> > >  At large N, $S_N(t)$ does not deviate significantly from its limit. This is what is referred to as consistency in Vyas et al, 2023.
> >
> > is a non-sequitur. By definition, for **any** $S_N(t)$ that has a limit it is true that $S_N(t)$ does not deviate significantly from the limit at large $N$.
> >
> > I guess a way to phrase this that would make more sense would be to just say:
> >
> > > $S_N(t)$ has a limit. This is what is referred to as consistency in Vyas et al, 2023.
> >
> > But at that point you might as well just say that $S_N(t)$ has a limit, rather than that it is consistent.
> >
> > That being said, I think informally referring to some quantities as being “consistent” (meaning they are either “close” to one another, or are “close” to some limit), like is done (Vyas et al. 2023) is perfectly clear, and doesn't need a formal or informal definition.
> >
> > The reminder of the changes also sound great. I would maybe slightly reword certain parts:
> >
> > > When $g(t)$ is an increasing function (up to fluctuations), Super Consistency is violated.
> >
> > nit: I would change this to “when $g(t)$ increases over time (up to fluctuations)...” just because “increasing function” is a mathematically commonly used term that implies monotonicity.
> >
> > ---
> > ### Clarifying the definition in the abstract:
> > Even given the clarification, I still don't think the line in the abstract is particularly clear:
> > > [...] **find that certain spectral properties under μP are largely independent of the width and depth of the network along the training trajectory. We name this property super consistency of the landscape.**
> >
> > I think a reader, after reading only the abstract, would have no idea that what you have in mind is what you later describe as super-consistency in Section 3. “properties being largely independent of size along the training trajectory” could just mean that they are consistent, or close to the limit at reasonable sizes. It's absolutely not clear that super-consistency encompasses whether the gap grows or shrinks as the training progresses.
> >
> > Here is a suggested alternative:
> > > we find that certain spectral properties under μP are largely independent of the width and depth of the network along the training trajectory, **and they become more consistent as the training progresses**. We name this property super consistency of the landscape.

---

> > > ### Comment · Reviewer_f2zq · 2024-08-09
> > > **Looking at the N-th largest eigenvalue or N-th percentile eigenvalue**
> > >
> > > ### Re: looking at the N-th largest eigenvalue or N-th percentile eigenvalue
> > > I realised after the rebuttal that looking at eigenvalues of the Hessian, although interesting, is kind of difficult to interpret, since the Hessian is composed of gradients for many parameters that grow at different rates with width (e.g. input layer vs middle layer weights).
> > >
> > > This is in contrast to hidden-layer weights in the infinite width limit. By RMT results, the per-layer eigenvalue (or singular value) spectrum of the weights should converge to a fixed spectrum in muP as width goes to infinity (and I think this holds throughout training). This means that top eigenvalue, but also something like the 10th percentile eigenvalue (or any other percentile) all have a well-defined limit.
> > >
> > > I think it would have been very interesting to see such results for changes in the weight eigenvalues and gradient eigenvalues on a per-layer basis, for different percentiles (i.e. comparing convergence for different parts of the limiting eigenvalue spectrum).
> > >
> > > This is a very late request, so I cannot ask the authors to implement it, and will not count this point in the final scoring. That being said, I'd personally be very much looking forward to seeing such experiments, and am disappointed I didn't notice the distinction sooner. I hope the authors consider running them for a camera-ready.

---

> > > > ### Comment · Reviewer_f2zq · 2024-08-09
> > > > **Response 3**
> > > >
> > > > I am looking forward to the authors' response. I am currently leaning towards increasing my score, but look forward to discussing the above points with the authors.

---

> > > > ### Author Response · Authors · 2024-08-10
> > > > **looking at the N-th largest eigenvalue or N-th percentile eigenvalue**
> > > >
> > > > This is a very interesting point and we thank the reviewer for bringing it up. Indeed it would be interesting to visualize the evolution of the whole spectrum (or some percentiles) at increasing $N$.
> > > >
> > > > > gradients for many parameters that grow at different rates with width
> > > >
> > > > In $\mu$P for gradient descent, the learning rate is set to $\eta_0 \gamma^2$, where $\gamma$ is $\mathcal{O}(\sqrt{N})$. This parametrization of the network is such that the magnitude of feature updates is $\mathcal{O}(1)$ for all layers. What does the reviewer mean that different weights move at different rates in $N$? Perhaps a simple example would help us understand.
> > > >
> > > > However, after the initial rebuttal phase, we thought about the convergence of the Hessian eigenspectrum in $N$ and its evolution during training. We set up the code to compute it and we are now in the process of estimating the spectral density and percentiles. We are partially confident that we will get these results before the end of the rebuttal and will keep the Reviewer updated on this matter. However, anonymous links are unfortunately not allowed and to our knowledge, we do not have a way to share the plots.
> > > >
> > > > We are also in the process of computing the per-layer Hessian. Unfortunately, we probably cannot deliver it by the end of the discussion phase, but will surely include it in the camera-ready version.
> > > > > “gradient eigenvalues on a per-layer basis”.
> > > >
> > > >  A clarification: what does the Reviewer mean by eigenvalues of a gradient here (which is a vector)?
> > > >
> > > > Finally, we would like to stress that the focus of our work is on the top $k$ eigenvalues (where $k$ is a fixed quantity equal to the number of classes) because in SGD the gradient lies in the top $k$ Hessian subspace, and on the sharpness because of its relevance in step size selection (and thus on learning rate transfer). We have also performed an experiment where we computed the curvature along the gradient direction (Figure 3 of the one-page pdf).

---

> > > ### Author Response · Authors · 2024-08-10
> > > **Answer to Response 1**
> > >
> > > We thank the reviewer for the additional valuable feedback. Indeed some of the extra points (e.g. convergence of the Hessian spectrum) have been in our minds after the first rebuttal and we have been thinking about that experiment.
> > >
> > > 1. **Clarifying the definition in the abstract**:
> > > By *certain spectral properties under μP are largely independent of the width and depth.* we mean that they are consistent (i.e. the finite width object is “close” to the infinite width one). And by *along the training trajectory* we mean that it remains independent of the size along the training trajectory (i.e. **super** consistent). However, we agree that the fact that we often observe the sharpness curves getting even closer to their large width limit is not stressed by this phrasing. Thus, we agree that “[...] become more consistent as the training progresses” better captures this intuitive meaning and have updated the abstract accordingly.
> > >
> > > 2. **On the meaning of “Consistency”**
> > > We partially disagree on the interpretation of “Consistency” in the work of Vyas et al (2023). The Reviewer (we apologize if we misunderstood) interprets consistency in a similar way as having a well-defined limit. In our interpretation, Consistency refers to the fact that at realistic widths (the word realistic is crucial here) the object of interest is (informally) practically converged to its limiting object. This is a very important result, as it implies that the infinite width model (proxied by a large width model in their work) is a good model for finite-width neural networks at realistic scales. In this sense, the existence of the limit alone does not imply consistency of the dynamics. In fact, under the NTK parametrization, consistency of the dynamics is not observed despite having a well-defined limit. Quoting from Vyas et al, 2023: “*We stress that this observed **consistency** is a property of networks in mean field/μP parameterization but is not present in other parameterizations which also give an infinite width limit like NTK parameterization*". We are slightly modifying our phrasing from "*At large $N$, $S_N(t)$ does not deviate significantly from its limit*" to "*At **realistically** large $N$, $S_N(t)$ does not deviate significantly from its limit*" to match the wording and meaning of consistency in Vyas et al (2023).
> > >
> > > Finally, we modified the phrasing of $g(t)$ increasing with time to the Reviewer’s suggestion of “when $g(t)$ increases over time (up to fluctuations)...”

---

> > > > ### Comment · Reviewer_f2zq · 2024-08-10
> > > > **Response**
> > > >
> > > > Thank you for the response, and being open to taking the feedback on-board.
> > > >
> > > > > We partially disagree on the interpretation of “Consistency” in the work of Vyas et al (2023). The Reviewer [...] interprets consistency in a similar way as having a well-defined limit. In our interpretation, Consistency refers to the fact that at realistic widths (the word realistic is crucial here) the object of interest is (informally) practically converged to its limiting object.
> > > >
> > > > Yes, I don't necessarily disagree, I think to a certain extent both meanings of consistency are used in Vyas et al. (2023). I think the  “at realistic scales” is not always implied by the term “consistency”, otherwise the “At Realistic Scales” in “Feature-Learning Networks Are Consistent Across Widths At Realistic Scales” would be superfluous in the title. I think sometimes in the text they just drop the “at realistic scales” and then ‘constistency’ implicitly captures that aspect as well. That being said, this current remark is pretty pedantic from my side, so I consider this settled.
> > > >
> > > > I agree with the authors that *“the word realistic is crucial”*, and its addition to their description of consistency makes it a non-trivial statement and a useful definition.
> > > >
> > > > Together with some extra clarification in the abstract (and hopefully similarly in the introduction wherever applicable) the authors mentioned above, I am happy to consider these points resolved.

---

> > > > > ### Comment · Reviewer_f2zq · 2024-08-10
> > > > > **Response 2**
> > > > >
> > > > > > In $\mu$P for gradient descent, the learning rate is set to $\eta_0 \gamma^2$, where $\gamma=\mathcal{O}(N^{0.5})$. This parametrization of the network is such that the magnitude of feature updates is order 1 for all layers.
> > > > >
> > > > > Depends on the implementation; when not using any multipliers (Table 3 implementation variant in [1]) the learning rates scale as $\Theta(N)$, $\Theta(1)$, or $\Theta(1/N)$ for input, hidden and output layers respectively.
> > > > >
> > > > > > What does the reviewer mean that different weights move at different rates in ? Perhaps a simple example would help us understand.
> > > > >
> > > > > Generally, since the changes input to any layer (activations/features) in $\mu$P will be order $\Theta(1)$, and there are $\Theta(N)$, that means that the change to the weights $\Delta W$ (resulting from a gradient step with a muP learning rate) must be roughly $\Theta(1/\sqrt{N})$ to ensure the output pre-activations are also $\Theta(1)$. However, this is not the case for the input layer; there are only $\Theta(1)$ inputs (rather than $\Theta(N)$) for the input layer. Hence, the size of the change $\Delta W$ to the input weights should also be order $\Theta(1)$ to ensure the entries of the output of that layer are $\Theta(1)$. That means there should be a discrepancy in the size of the changes $\Delta W$ to the weight matrices in the hidden and last layer ($\Theta(1/\sqrt{N})$) and input matrices $\Theta(1)$.
> > > > >
> > > > > Similar reasoning applies to biases, where the change after a gradient update $\Delta b$ will generally be $\Theta(1)$ as well.
> > > > >
> > > > > Of course, the above applies to changes to weights and biases (i.e. gradients multiplied by per-layer learning rates), and not to the gradients (which haven't been multiplied by a learning rate yet). In general, these will also have different scales for different layers (input, hidden, last).
> > > > >
> > > > > Now, this becomes doubly weird when considering something like the gradient vector or Hessian matrix for the entire network. There, you will have elements that grow at different rates with width (depending on whether they are weights or biases for an input, hidden or last layer), and also the *proportion* the elements will be different for different layers. For example, the number of the elements in the gradient vector corresponding to weights of a hidden layer grows as $N^2$, but for the input and last layers the number of elements grows as $N^1$. Hence, for large enough widths almost all elements of a gradient vector or Hessian matrix will correspond to hidden layer weights.
> > > > >
> > > > > ---
> > > > > > A clarification: what does the Reviewer mean by eigenvalues of a gradient here (which is a vector)?
> > > > >
> > > > > I mean the gradient $\frac{\partial L}{\partial W^{(\ell)}}$ for a particular weight matrix $W^{(\ell)}$, reshaped to have the same shape as that weight matrix. If it's not square, I mean the singular values.
> > > > >
> > > > > To be more specific, I did actually have the weight update $\Delta W^{(\ell)}$ in mind (i.e. the gradient multiplied by the learning rate in the case of SGD), since I think in that case, the $\mathcal{\Theta}(1/\sqrt{N})$ scaling for hidden layer weights means that the spectrum should converge to something well-defined, and for others it might not (although I'm not 100\% sure about this claim, I haven't worked through the maths in detail).
> > > > > ### muP check
> > > > > As an aside, did the authors do a muP coordinate check, as recommended in Appendix D.1 and Figure 5 of [1]? Speaking from personal experience, it's quite easy to mess-up the implementation of muP as there are many small caveats to the general rules (e.g. how to initialise LayerNorm parameters), and sometimes one can get learning rate transfer even when there are minor mistakes in the implementation. The coordinate checks usually reveal those. I think adding these to an appendix would make the authors' empirical claims significantly more resistant to any potential readers' doubts.
> > > > >
> > > > > [1] Tensor Programs V:  Tuning Large Neural Networks via Zero-Shot Hyperparameter Transfer

---

> > > > > > ### Author Response · Authors · 2024-08-10
> > > > > > **Response**
> > > > > >
> > > > > > **On the definition of Super Consistency**
> > > > > >
> > > > > > We are glad that the issue of the definition (which was the main issue raised in the initial review) is resolved, and we sincerely thank the Reviewer for the feedback, which has ultimately helped our paper. We will make it in clear what we mean by Super Consistency whenever it applies.
> > > > > >
> > > > > > **Equivalent parameterizations of $\mu$P**
> > > > > > > Depends on the implementation; when not using any multipliers (Table 3 implementation variant in [1]) the learning rates scale as …
> > > > > >
> > > > > > We haven’t included the experiments, but as a sanity check, we have tried to repeat some experiments with the code of Yang et al [1] (https://github.com/microsoft/mup), which adopts a different parametrization (which is equivalent in terms of the predictor’s dynamics). In this case, instead of computing the largest eigenvalue of $\gamma^2 N$, we have computed the largest eigenvalue of $DH$, where $D$ is a diagonal matrix containing the width-dependence of the learning rate for each parameter. In both $\mu$P parametrizations, we get super consistency of the sharpness, converging at the EoS value of $2/\eta_0$. Thus, the largest eigenvalue stays the same regardless of the (rescaled) Hessian. This is as expected, as the sharpness should be invariant to the equivalent $\mu$P reparametrizations. This also makes us confident about our implementation.
> > > > > >
> > > > > > Furthermore, in the experiments of the attached one-page pdf, we also use the code in https://github.com/microsoft/mup, and compute the largest eigenvalue of $DP^{-1}H$, where $P$ is the diagonal preconditioner computed by Adam (calculated as in the AEoS paper of Cohen et al (2022)). We have added a section in the appendix describing the Hessian computation for Adam + width scaling.
> > > > > >
> > > > > > > change to the weights $\Delta W$ is different for different layers
> > > > > >
> > > > > > and
> > > > > >
> > > > > > > Now, this becomes doubly weird when considering something like the gradient vector or Hessian matrix for the entire network. There, you will have elements that grow at different rates with width …
> > > > > >
> > > > > > We think the above discussion answers this question, i.e., we observe Super Consistency of the Hessian’s largest eigenvalue when the Hessian is rescaled appropriately to account for the width dependence of the learning rates and preconditioner (as a subtlety, notice that with the same meaning of $P$ as above, $P^{-1/2}HP^{-1/2}$ has the same eigenvalues as $P^{-1}H$ (Cohen et al (2022))). We will make more clear in the camera-ready version.
> > > > > >
> > > > > > > spectrum of $\Delta W^{\ell}$
> > > > > >
> > > > > >  It would be of course of crucial interest to derive the eigenvalue (or singular value) distribution of something like $\Delta W^{\ell}$ under $\mu$P. However, we hope that the Reviewer agrees that this is related but slightly beyond the scope of the paper. The evolution of the spectrum of the weights through training has been empirically analyzed in previous work aiming for “effective theories” of neural network training (e.g. Martin and Mahoney https://www.jmlr.org/papers/volume22/20-410/20-410.pdf). There, it is observed that the spectrum of correlation matrices of the weight can exhibit spikes, which can be modeled by spiked covariance models in RMT.
> > > > > >
> > > > > > **Implementation of $\mu$P**
> > > > > >
> > > > > > Our results are fully compatible with the existing open-source implementation of $\mu$P (https://github.com/microsoft/mup). We explicitly test this in a subset of experiments. Furthermore, the new rebuttal-time experiments of Adam + width scaling use this codebase. Also, we test networks up to very large widths and depths (up to 300 million parameters), which makes us confident that our implementation is correct. For the layer norm parameters, we follow the prescription on page 24 of Yang and Hu (https://arxiv.org/pdf/2203.03466). We will make the code available if/when the paper is accepted.
> > > > > >
> > > > > > We hope that this resolves these issues, and we are open to further discussion.

---

> > > > > > ### Author Response · Authors · 2024-08-11
> > > > > > **Response**
> > > > > >
> > > > > > As the end of the rebuttal period is approaching, we would like to once again thank the reviewer for their valuable feedback and time, and ask again if there are any other points that the reviewer would like to discuss.
> > > > > > Otherwise, if we have settled all the open points (especially surrounding the main initial issues raised by the reviewer regarding the definition, which have been resolved and clarified with the reviewer’s valuable feedback) we kindly ask the reviewer to revise their original score.

---

> > > > > > > ### Comment · Reviewer_f2zq · 2024-08-13
> > > > > > >
> > > > > > > After the discussion with the authors and agreeing on changes to be made to the paper, I increased my score and would recommend acceptance.

---

> > > > > > > > ### Comment · Reviewer_f2zq · 2024-08-13
> > > > > > > >
> > > > > > > > As an aside, I would still implore the authors do run a muP coordinate check on their implementation before writing up a camera-ready. I would feel much more comfortable having recommended the acceptance of this paper knowing this unit-test passed.

---

> ### Author Response · Authors · 2024-08-14
>
> We deeply appreciate that the reviewer is now in favor of acceptance.
>
> To wrap it up, we will include the additional experiments on the hessian's eigenspectrum, and clarify the meaning of Super Consistency at all points in the paper according to the meaning and phrasing agreed here. We will also run a coord check experiment to verify that features updates are $\mathcal{\Theta}(1)$.

---

### Author Rebuttal · Authors · 2024-08-07

We thank the reviewers for their initial reviews and interesting comments. In particular, we summarize that all the reviewers have acknowledged the validity and importance of Super Consistency as a novel and important property for understanding the loss landscape of neural networks at different scales, in particular with relation to the phenomenon of learning rate transfer.

We did not find any common weakness shared across the majority of reviewers. The main concerns are: Super Consistency under very large models (Rev. 47kG and ssWb) and other optimizers such as AdamW (Rev. 11Ly) and the definition of the term Super Consistency (Rev. F2zq).

1. **On larger scale experiments**. We would like to note that the main published work in this area that we take as reference (Cohen et al, 2021, Gur-Ari et al, 2018) train smaller models, and on a less diverse set of tasks and architectures. In fact, we have experimented across several tasks (CIFAR-10, Imagenet (vision), and wikitext (language)) with two different architectures: ResNets and Transformers (including Vision Transformers). We have also performed experiments on several parametrizations, including NTK, $\mu$P and SP in width, and Depth-$\mu$P and residual networks without $1\sqrt{L}$ scaling for depth. We have trained models with up to 96 layers in depth (e.g. Figure 19), and 8172 units in width. Thus, we believe that our claims remain valid to a sufficient scale, that is compatible with previous work in this area. However, it remains interesting to see what would happen at an even larger scale. Thus, **we ran new experiments scaling up to about 300 million parameters**. See next:
2. **New experiments (Figures refer to the one-page pdf)**:
    + **Adam + Post-LN Transformers, width scaling (Figure 1)**. We have adapted the Hessian computation to the Adam case as in
            Cohen et al. (https://arxiv.org/pdf/2207.14484) (Eq. 2), where the largest eigenvalue of the preconditioned Hessian is computed:
            $\lambda_{\max}(P^{-1}H)$, where P is the diagonal Adam’s preconditioner. This is important for width scaling (in contrast to
            depth scaling) because the preconditioner itself is width-dependent. Results are in the attached pdf (Figure 1). We show that
            Super Consistency holds for a Post-LN Transformer architecture trained on wikitext, where the largest network has about 300
            million parameters.
    + **Alignment of gradient and top hessian eigenvector (Figure 3)**. Rev. 47Kg has raised a concern about how the Super
            Consistency of the first few eigenvalues can explain learning rate transfer. The fact that only the first eigenvalues are important is
            a finding of Gur-Ari et al, 2018, where it is shown that SGD happens in the space spanned by the top $k$ Hessian eigenvectors.
            However, we have performed an experiment where we compute $g^T H g / ||g||^2$, which is an unnormalized measure of
            alignment between gradient and Hessian, thus capturing the curvature along the gradient direction. We observe Super
            Consistency in two experiments using residual networks, both under $\mu$P (width scaling) and Depth-$\mu$P (depth scaling).
            This further strengthens the connection between the Super Consistency of the landscape and learning rate transfer and we
            believe it resolves the reviewer’s concern.
    + **Adam and AdamW experiments, width scaling (Figure 2)**.  We have performed new width-scaling experiments with Adam
               and AdamW on 3 layers Convolutional Networks, extending Super Consistency to these settings.

3. **Definition of Super Consistency**. Rev. F2zq has expressed concerns about Definition 3.1. We would like to point out that we found it overcomplicating to aim for an airtight mathematical description of our observation. To enhance clarity, we opted instead for an actionable definition that would reflect the actual measurements of e.g. Figure 3. However, we agree with the reviewer that some parts of Definition 3.1 can be potentially confusing, and can be interpreted as an attempt of rigorous formalization. We provide a revision below which considers all of the Reviewer's helpful comments. We are sorry the initial writing caused confusion, and we are positive our current description, now under no claim of "definition", better reflects our mental picture and resolves potential doubts. We have updated the paper by removing the Defintion environment and partially rephrasing some parts. Please see the rebuttal to Rev. F2zq for details.

---

### Decision · Program_Chairs · 2024-09-25

**Decision:**

Accept (poster)

**Comment:**

The paper investigates the sharpness dynamics, the training loss dynamics and the learning rate transfer under $\mu$P, and explain why there is a consistency (named Super Consistency) of some properties across different model sizes.

The reviewers discussed the paper in detail (regarding the definition of the term "Super Consistency", large-scale experiments, experiments with Transformers, theoretical limitations beyond 2-layer NNs, etc.) and agreed upon positive ratings. I recommend accept.
I hope the authors incorporate the discussion with the reviewers into the camera-ready version.